# Automated single-molecule imaging in living cells

Masato Yasui[1], Michio Hiroshima[1,2], Jun Kozuka[1], Yasushi Sako[2] & Masahiro Ueda[1,3]

An automated single-molecule imaging system developed for live-cell analyses based on artificial intelligence-assisted microscopy is presented. All significant procedures, i.e., searching for cells suitable for observation, detecting in-focus positions, and performing image acquisition and single-molecule tracking, are fully automated, and numerous highly accurate, efficient, and reproducible single-molecule imaging experiments in living cells can be performed. Here, the apparatus is applied for single-molecule imaging and analysis of epidermal growth factor receptors (EGFRs) in 1600 cells in a 96-well plate within 1 day. Changes in the lateral mobility of EGFRs on the plasma membrane in response to various ligands and drug concentrations are clearly detected in individual cells, and several dynamic and pharmacological parameters are determined, including the diffusion coefficient, oligomer size, and half-maximal effective concentration ($EC_{50}$). Automated single-molecule imaging for systematic cell signaling analyses is feasible and can be applied to single-molecule screening, thus extensively contributing to biological and pharmacological research.

[1] Laboratory for Cell Signaling Dynamics, RIKEN BDR, 6-2-3, Furuedai, Suita, Osaka 565-0874, Japan. [2] Cellular Informatics Laboratory, RIKEN, 2-1 Hirosawa, Wako 351-198, Japan. [3] Laboratory of Single Molecule Biology, Graduate School of Frontier Biosciences, Osaka University, 1-3 Yamadaoka, Suita, Osaka 565-0871, Japan. These authors contributed equally: Masato Yasui, Michio Hiroshima. Correspondence and requests for materials should be addressed to Y.S. (email: sako@riken.jp) or to M.U. (email: masahiroueda@riken.jp)

Single-molecule imaging of biomolecules in living cells allows for the investigation of cell signaling and other molecular mechanisms[1–3]. These techniques have enabled direct monitoring of the behaviors of biomolecules in living cells and the quantitative detection of the locations, movements, turnovers, and complex formations of biomolecules with single-molecule sensitivity; thus, these techniques represent powerful tools that can be used to elucidate the molecular mechanisms underlying intracellular signaling processes. Systematic and comprehensive measurements of numerous molecular species with single-molecule sensitivity provide detailed information regarding elementary biological processes and new insights into system dynamics[4], thereby deepening and extending current biological and medical knowledge. However, the techniques used to date in large-scale experiments to investigate various types of molecular/cellular/drug species under constant and well-controlled experimental conditions have not reached the single-molecule level in living cells. Significant expertise is needed for focusing at nanometer precision, searching for cells suitable for observation, and statistically analyzing individual molecules, and the lack of such skills prevents time-efficient and nonbiased mass data acquisition and analysis. Therefore, we developed a fully automated in-cell single-molecule imaging system (AiSIS) based on an artificial intelligence-assisted total internal reflection fluorescence microscope (TIRFM), which has the potential to pave the way for the widespread use of single-molecule imaging technology in the biological and medical sciences. The apparatus dramatically reduces the time required for imaging and analysis by ~10-fold for researchers familiar with single-molecule measurements. For researchers who are not familiar with the method, AiSIS might eliminate the need to learn the method and reduce the time requirements by a factor of more than 100. Moreover, the newly developed elementary techniques equipped in AiSIS can be applied to general high-magnification microscopy to automate conventional routines, thereby dramatically improving the current situation of imaging and analysis in life science studies, which currently requires considerable time and effort.

## Results

**Automated large-scale single-molecule imaging.** Figure 1a presents an illustration of AiSIS. TIRF optics and a robotized manipulator were constructed in an incubation chamber used for cell culture (IMACS, Hamamatsu) to maintain cellular physiological conditions under constant temperatures and water vapor and $CO_2$ concentrations (also see Supplementary Figure 1a). We used multi-well plates (typically 96 wells) to sequentially observe multiple samples under different experimental conditions. Supplementary Movie 1 demonstrates the procedure for the automatic measurement. Figure 1b and Supplementary Movie 2 show single-molecule images of GFP-labeled epidermal growth factor receptors[5] (EGFR-GFPs) expressed in the plasma membrane of CHO-K1 cells. Observations of five cells before and after stimulation with 60 nM EGF or mock solutions in 60 different wells (a total of 600 cells) were performed within 8 h and 30 min (510 min) (see below for details). We confirmed that 591 of the 600 cells were successfully recorded for further statistical analysis. The remaining nine cells were excluded because the single-molecule tracking software failed to continuously track any fluorescent spots for more than 1 s.

The key techniques used to achieve automation in addition to the TIRF optics for single-molecule imaging[6] included immersion-oil feeding, autofocusing, and auto-cell searching (Fig. 1c–e), and they rendered the system applicable to many types of microscopy in addition to TIRFM. The automatic oil reflux system maintains the oil volume at the objective constant,

thus enabling long-term observations over one day (Fig. 1c). An objective lens adaptor with two outlets for discarding oil was effective in preventing the oil in the space between the objective and the bottom glass plate from moving a long distance (see the Methods section). The oil reflux system is widely applicable to other optics (e.g., confocal, differential interference contrast (DIC), etc.) with a high-magnification (e.g., ×100) and large N.A. (e.g., 1.49) lens. The focusing system, which can automatically set the objective to the in-focus position (see the Methods section for the algorithm), consists of a newly developed optical system and image processing algorithm that includes deep learning[7] (Supplementary Movie 3). The detailed algorithm is explained in the Methods section. In brief, the algorithm refers to an image of the iris, which is located at the optically conjugate plane to the upper surface of the cover slip, taken using a surface reflection interference contrast (SRIC) filter (Fig. 1d and Supplementary Figure 1b). A shift in the glass surface blurs the image of the iris when the image is out of focus. The in- and out-of-focus images of the iris (400 images) were prepared as training data for deep learning and used to train the neural network. During coarse shifting of the objective position, the trained neural network judged the iris images to determine whether the iris was in focus or not. After this coarse focusing, the sharpness of the iris image was evaluated to obtain the precise focus position. The two-step focusing procedure has the advantages of both deep learning-based prompt determination and precise image processing-based determination of the in-focus position. The in-focus position was visually assessed and varied within a standard deviation of 181 nm, which was sufficient for single-molecule imaging (Fig. 1d). We succeeded in perfectly autofocusing the glass surface in 100 trials (Supplementary Figure 1c).

The deep-learning methods also provide an auto-cell searching algorithm without the need to manually design imaging filters for cell selection (Fig. 1e and Supplementary Figure 2). In single-molecule imaging analyses of fluorescently labeled molecules expressed in living cells, the density of the fluorescent spots should be $1–3\ \mu m^{-2}$ in the field of view at a spatial resolution of ~250 nm. Because the expression level of fluorescent proteins usually varies by a factor of 10 from cell to cell, only cells with suitable spot densities are selected. To identify these cells, the fluorescent spot images were acquired using serial X–Y scanning (Supplementary Movie 4) and then regions similar to previously learned images with suitable spot densities were selected (Fig. 1e, upper right). Since out-of-focus images can also be trained, our system can be applied to slightly blurred images, which are frequently observed during scanning to select cells with an adequate density of fluorescent spots. The required number of training images for successful automatic cell searching was at least 40 (see the Methods section for details). In addition, we applied deep-learning methods to the SRIC images of the cells to obtain the actual cell regions (Fig. 1e, lower right), which should be clearly distinguished from regions outside the cells in which bright spots caused by fluorescent debris often occur. For successful cell region detection, ~200 training images were required for the learning procedure. The networks trained using CHO or HeLa cells could be practically used for recognition of HeLa or CHO, respectively (see the Methods for details), indicating that additional training is not required for each cell type if the cell images exhibit similar visual features. However, if the CHO cells must be distinguished from HeLa cells, then the neural network must learn another specific feature that obviously differs between these cells.

**Automatic detection of EGFR changes upon ligand stimulation.** We assessed the capacity and efficiency of the automated

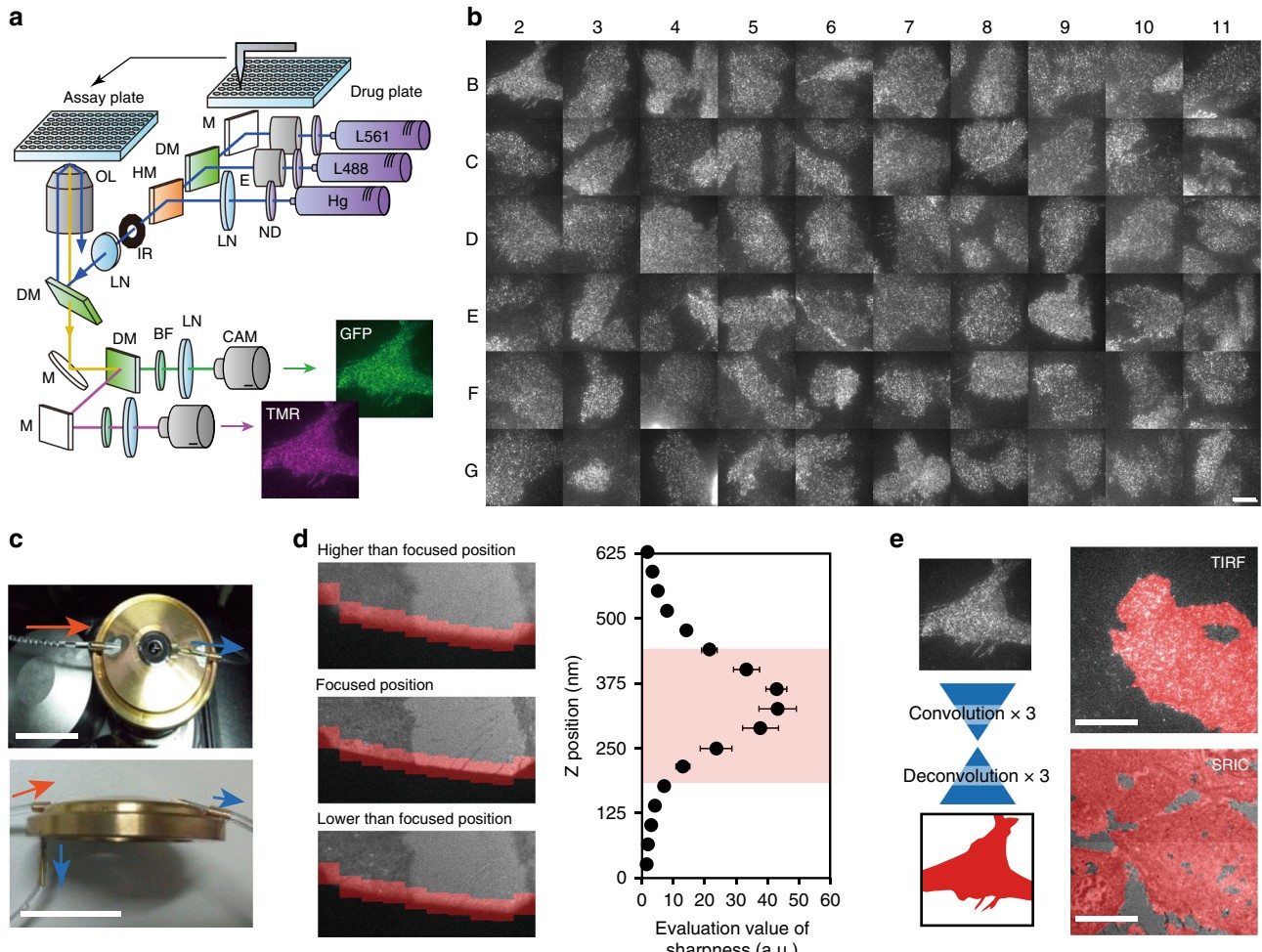

**Fig. 1** Automated single-molecule imaging system. **a**, Schematic diagram of the system. L488 and L561 are lasers with wavelengths of 488 and 561 nm, respectively; Hg mercury lamp, OL objective lens, LN lens, DM dichroic mirror, M mirror, ND neutral density filter, IR iris, E beam expander, BF bandpass filter, CAM EMCCD camera. **b** Single-molecule images of EGFR-GFP in CHO-K1 cells in 60 wells of a 96-well plate (none in the peripheral wells). Alphanumeric characters indicate the well number. Scale bar: 10 μm. **c** Immersion-oil feeding system. Oil flows into the adaptor to the objective lens (upper: top view, lower: side view. A cartoon structure is shown in Supplementary Figure 9) through one inlet in the direction of the red arrow and out two outlets along the blue arrows. Scale bars: 20 mm. **d** Autofocusing algorithm. Conjugated images of the iris and basal surfaces of the cells are shown for the in-focus and out-of-focus positions. Pixel intensities in the red squares on the edge of the iris are measured to calculate the evaluation values plotted in the right panel. A red background indicates Z positions observable in a nonblurry image of the fluorescent spots. Each point consists of 5 data points. Error bars: SD. **e** Deep learning-based cell searching method consisting of three convolution and three deconvolution processes (left). Parameters of the neural network were adjusted by learning using training data. Cells with suitable expression of fluorescent molecules (right, upper) and cell regions in the SRIC image (right, lower) are shown in red. Scale bars: 10 μm

apparatus by performing a single-molecule analysis of the mobility and signaling of EGFRs in CHO-K1 cells. The EGFR is a transmembrane receptor tyrosine kinase that transmits extracellular signals to the cytoplasm via the EGF-RAS-MAPK cascade[8,9]. Overexpression of EGFRs is typically observed in various cancer cells, and EGFR mutations lead to constitutive signaling. Due to these characteristics, the EGFR is an attractive target for cancer therapy[10–12]. Several previous studies have performed single-molecule imaging of EGFRs and reported that EGF can induce EGFR phosphorylation, which slows EGFR mobility within a confined area on the membrane, a phenomenon accompanied by EGFR clustering[13–17]. EGF clusters larger than a dimer have been suggested to be significant for downstream signaling[17]. Initially, we performed a simple assessment to distinguish between cells with and without EGF addition by detecting mobility changes in the EGFR. CHO-K1 cells expressing EGFR-GFP were observed sequentially at the basal membrane in 60 wells (Fig. 1b and Supplementary Movie 2). Figure 2a shows

the time evolution of the mean square displacement (MSD), which is an index of mobility, obtained from 60 different wells. No obvious changes were observed in the cells exposed to the mock solution, whereas the cells exposed to EGF exhibited changes in the MSD and the confinement length was shortened after EGF stimulation. The statistical analysis performed by AiSIS confirmed with reproducibility the difference in behavior between the cells exposed to EGF and the mock solution.

Using vast amounts of single-molecule tracking data, we can characterize the molecular properties of EGFRs with low measurement errors by determining certain parameters, such as the diffusion coefficient and oligomer size. As shown in Fig. 2a, ~236,035 EGFR-GFP spots from ~591 cells (60 wells) were imaged and analyzed within 8 h and 30 min. Figure 2b shows the average MSD of the EGFRs calculated from tracking data obtained from wells containing EGF or mock solution. The distributions of the diffusion coefficients were also calculated using the same data (Fig. 2c and Supplementary Figure 3a). The

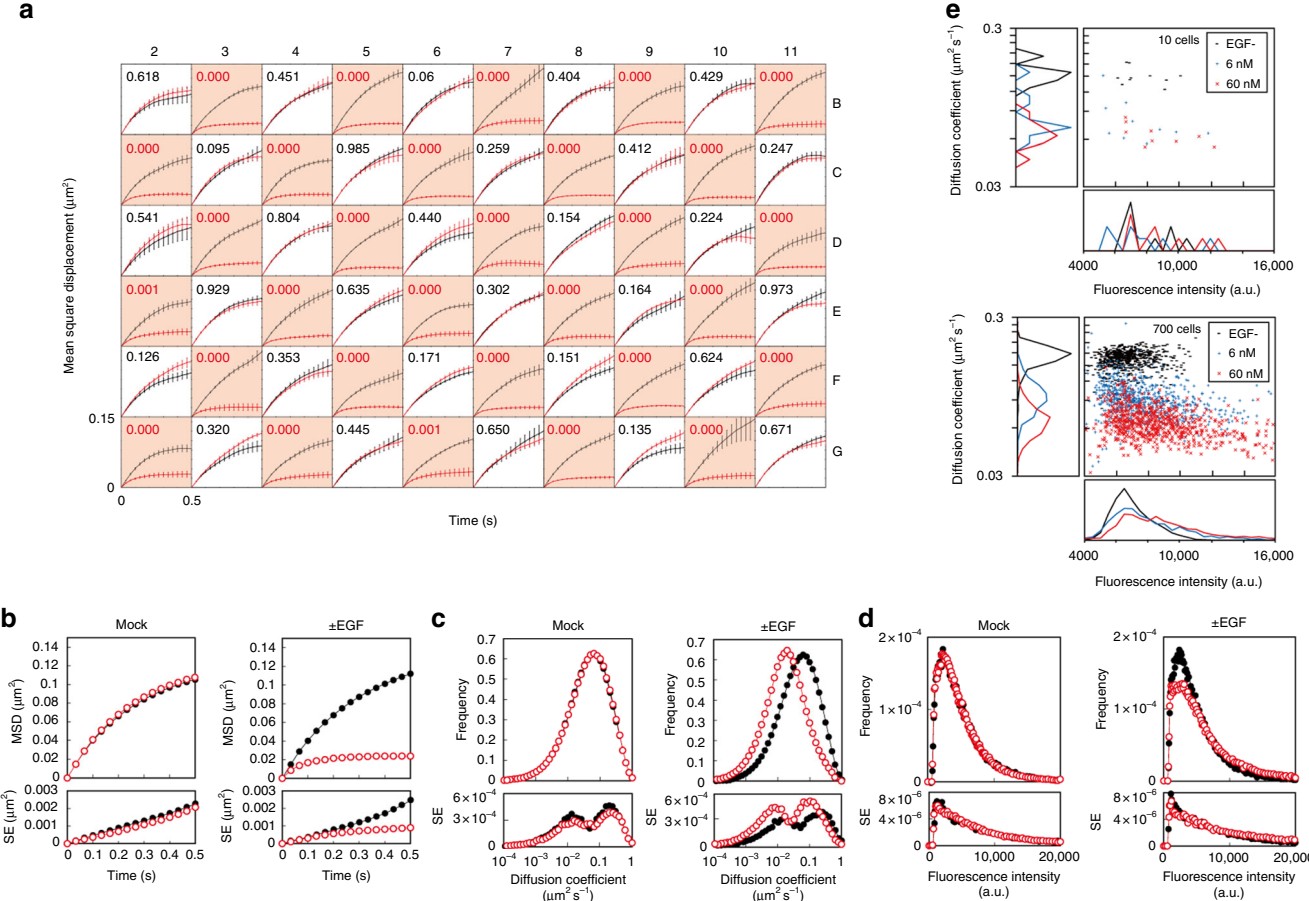

**Fig. 2** Single-molecule analysis in living cells using the automated system. **a** Time evolution of the MSD of EGFR-GFP spots analyzed using the data shown in Fig. 1b. EGF (60 nM) or mock solution was added to each well (red or white background, respectively). *P*-values of the distribution of the MSD at $\Delta t =$ 66 ms before and after addition of the solutions are indicated, and significant differences (<0.005) are shown in red. The number of observed cells and molecules was 591 and 236,035, respectively. The MSD was calculated for every cell. Curves and error bars indicate the average and SE between cells. **b–d** Changes in the behavioral properties of EGFR molecules induced by EGF. Upper and lower panels represent the mean and SE between cells, respectively. A recording of a cell was considered successful if the number of spots in a 1 s observation was greater than zero. Distributions of the MSD (**b**), diffusion constant at $\Delta t = 66$ ms (**c**), and fluorescence intensity in the first frame (**d**) are shown before (black) and after (red) the addition of EGF (right) or mock (left) solution. **e** Distribution of diffusion coefficients and fluorescence intensities in 10 (upper) and 700 (bottom) cells. The means and standard deviations at 0, 6, and 60 nM EGF were 0.16 ± 0.022, 0.092 ± 0.027, and 0.065 ± 0.017 $\mu m^2 s^{-1}$ for the diffusion coefficients and 6766 ± 1229, 7790 ± 2576, and 8849 ± 3393 a.u. for the fluorescence intensities, respectively. Panels on the left and bottom of each two-dimensional map show the histograms of the diffusion coefficients and fluorescence intensities, respectively

diffusion coefficients averaged over all observed cells showed a decrease upon EGFR activation from 0.11 ± 0.01 (148 cells) to 0.051 ± 0.008 $\mu m^2 s^{-1}$ (149 cells), which was expressed as the average ± SE. The MSD curve in Fig. 2b shows a shortening of the diffusion area of the EGFR upon activation, with the length confined[18] to 264 nm on average over all cells (see Eq. (7) in the Methods section for analysis). The fluorescence intensity distributions of individual spots reflect oligomerization of the EGFR (Fig. 2d and Supplementary Figure 3b). The estimated oligomer sizes per fluorescent spot were 1.5 ± 0.9 and 2.1 ± 1.7 on average ± SD before and after EGF binding, respectively (Supplementary Figure 3). The data quality was improved by increasing the number of molecules analyzed as shown in Supplementary Figure 5. Although several hundred cells can also be observed and analyzed manually by human experimenters, a considerably longer amount of time, i.e., occasionally a week or longer, is required depending on the proficiency of the experimenters, and the data could suffer from human errors and biases. In contrast, AiSIS can easily perform large-scale imaging analyses. Figure 2e shows a typical example of the EGFR mobility shift and

oligomerization with cell-to-cell heterogeneity (See also Supplementary Figure 6) from 2100 cells at various EGF concentrations. Such cellular heterogeneity could affect the reproducibility of the data if only a small number of cells is analyzed.

**Acquisition of kinetic and pharmacological parameters**. Using multi-well plates, multiple samples can be observed under different experimental conditions. Elucidating the effects of ligands, such as agonists, antagonists, and inhibitors, on target molecules has important implications in biology and pharmacology. We performed an automatic analysis of 10 different EGF concentrations. The MSD values at $\Delta t = 66$ ms ($MSD_{\Delta t = 66 ms}$) were calculated under each condition. We could fit the data to a biphasic curve at a half-maximal effective concentration ($EC_{50}$) of 6.6 nM and a Hill coefficient of 1.0 (Fig. 3a). The $EC_{50}$ and Hill coefficient were consistent with those described in previous reports, confirming that the EGFR has several affinities in ligand kinetics[19] and suggesting that the $EC_{50}$ primarily reflects the lower-affinity site. Furthermore, AiSIS can assess complex stimulations of ligands and inhibitors at various concentrations. EGFR

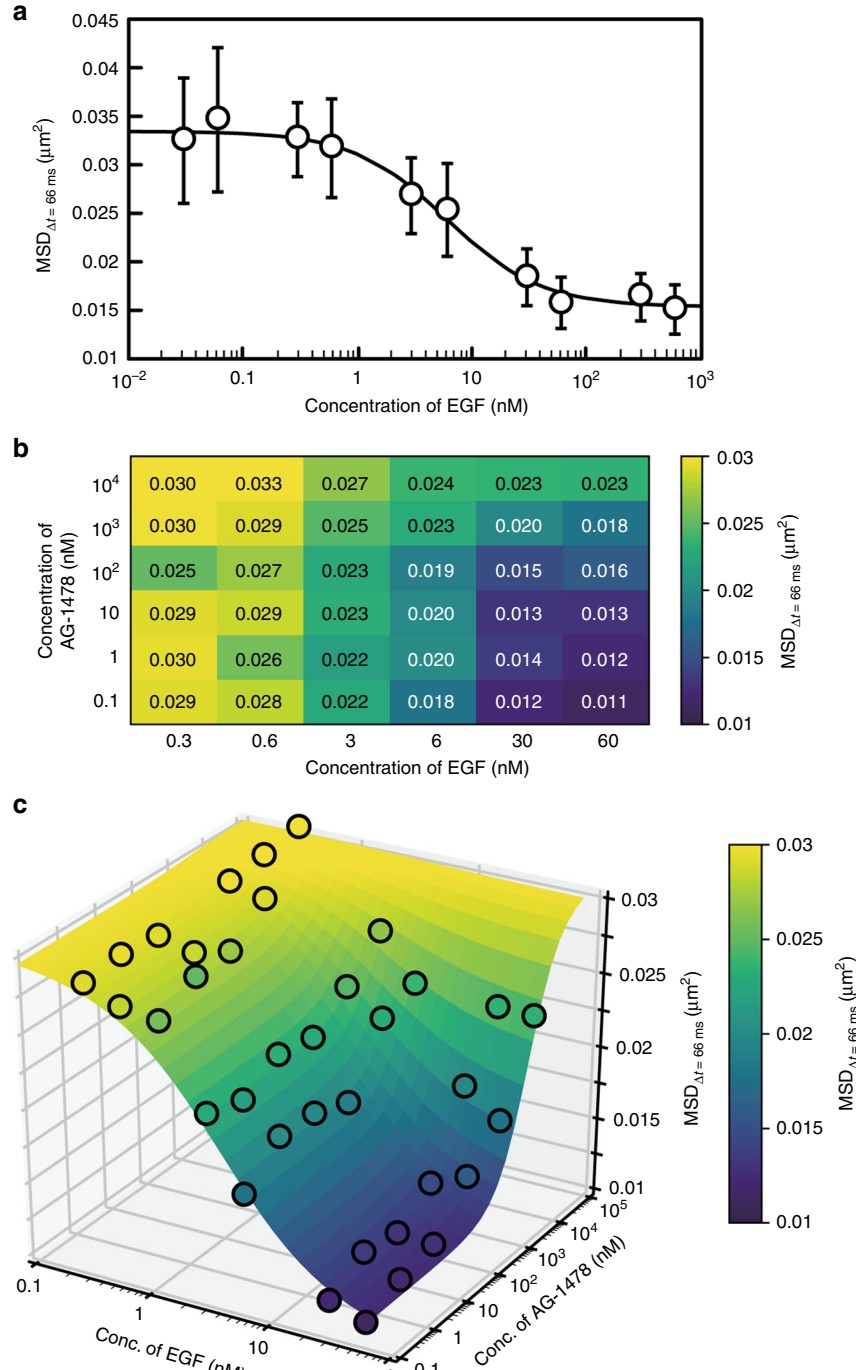

**Fig. 3** Pharmacological parameters obtained by the automated apparatus. **a** Dose–response curve of the MSD at $\Delta t = 66$ ms showing clear dependence on the EGF concentration. In total, 18 cells were analyzed for each data point. Error bars: SD. The curve was calculated from Eq. (14), and the obtained parameters are shown in Supplementary Table 4. **b**, **c** Antagonistic effects of EGF and the inhibitor (AG1478) on the MSD at $\Delta t = 66$ ms of EGFR mapped in two-dimensional (**b**) and three-dimensional (**c**) space. Colors correspond to the MSD at $\Delta t = 66$ ms according to the color bar. Dots and the surface indicate the measured data and fitted curve calculated from Eq. (16), respectively. The obtained parameters are shown in Supplementary Table 5. In total, three cells were used for each data point

phosphorylation is known to be inhibited by tyrphostin AG1478, which binds to the ATP binding site at the cytoplasmic region of the EGFR. Thus, AG1478 suppresses the proliferation of endometrial and ovarian cancer cells[20]. All combinations of the six concentrations of EGF and AG1478 (a total of 36 conditions) were automatically measured in one experiment using a 96-well plate. The observed MSD values at $\Delta t = 66$ ms were mapped in a two-dimensional space (Fig. 3b). AG1478 clearly suppressed the

EGF-dependent decrease in the MSD value in a manner that resembled the effect of receptor phosphorylation, thus indicating an antagonistic effect on the receptor mobility. By fitting the data using Eq. (15), which considers the noncompetitive inhibition scheme of the EGFR, the dissociation and inhibitory constants were calculated as $EC_{50}$ ($K_D$) = 4.7 nM and $IC_{50}$ ($K_i$) = 2.3 µM for EGF and AG1478, respectively (Fig. 3c), which are consistent with previously reported values[20,21].

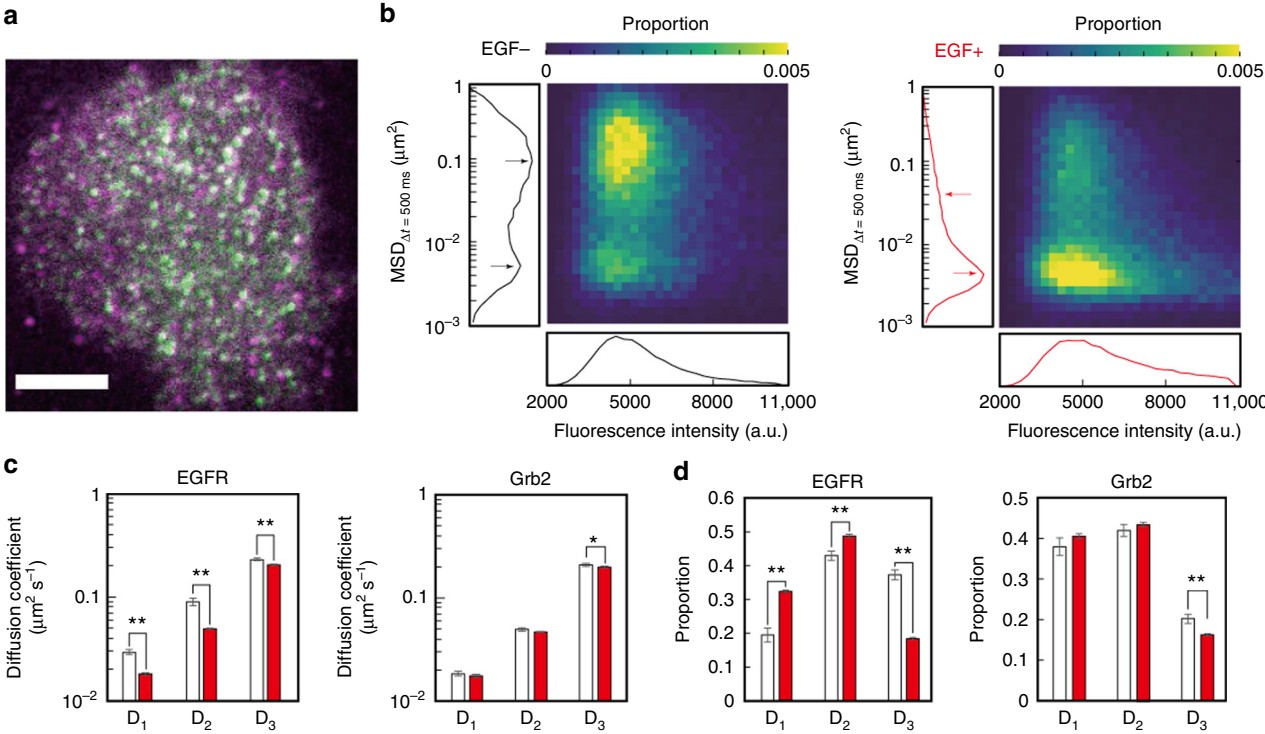

**Fig. 4** Two-color automated single-molecule imaging. **a** Merged image of EGFR-GFP (green) and Grb2-TMR (magenta) at the basal cell surface. Magenta spots shown outside the cell are free TMR molecules absorbed on the coverslip. Scale bar: 5 μm. **b** Heat maps of the MSD at 500 ms and fluorescence intensities before (left) and after (right) 60 nM EGF stimulation. Panels on the left and bottom of each heat map show histograms of the MSD at 500 ms and fluorescence intensities, respectively. Arrows indicate the peaks of subpopulations corresponding to the EGFR states. The number of observed cells before and after EGF addition was 148 and 146, respectively. **c**, **d** Averaged diffusion coefficients (**c**) and fractions (**d**) of three EGFR (left) and Grb2 (right) motional states before (white) and after the addition of EGF (red). Asterisks indicate significant differences (*$P < 0.05$, **$P < 0.01$). The number of cells before and after the addition of EGF was 4 and 22 in the EGFR observation and 44 and 20 in the Grb2 observation, respectively. Error bars: SE

## Analysis of signal transduction by EGFR in living cells.

Our apparatus also allows for the simultaneous observations of two types of molecules labeled with different colors. Upon EGF stimulation, Grb2 binds phosphorylated EGFR and activates the signaling pathway[22,23]. To observe signal transduction, we simultaneously observed EGFR-GFP and Grb2 tagged with HaloTag and stained with tetramethylrhodamine (Grb2-TMR) (Fig. 4a and Supplementary Movie 5). Heat maps based on the MSD at $\Delta t = 500$ ms and the fluorescence intensities of the individual EGFR-GFP molecules showed multiple states along with transitions to slower and brighter subpopulations upon EGF stimulation (Fig. 4b). The multiple states of EGFR and Grb2 were characterized based on the diffusion coefficients using previously reported methods[24–26]. The minimum state numbers were determined using the Akaike information criterion (AIC) (see the Methods section for details). Both EGFR and Grb2 were likely to adopt three motional states regardless of EGF stimulation (Supplementary Figure 7), which were named immobile, slow-mobile, and fast-mobile states based on the obtained diffusion coefficients. As shown in Fig. 4b, the MSD at a long duration (500 ms) clearly differed between the fast-mobile state and the two slower states, but the measurement was not sufficient to resolve the slow-mobile and immobile states due to the substantially broad distribution of the displacement. EGF caused obvious changes in the diffusion coefficients and fractions of the EGFR mobility states, and the immobile and slow-mobile fractions increased while the fast-mobile fraction decreased (Fig. 4c, d and Supplementary Figure 8a).

Furthermore, measurements at multiple time points provided the time course of the state transitions and the fluorescence intensity distribution. After EGF addition at 33 s, the slow-mobile fraction of the EGFR reached a maximum and the fast-mobile fraction decreased. At 89 s, the immobile state reached its peak (Fig. 5a and Supplementary Figure 8b). The fluorescence intensity, which reflects receptor clustering, reached a maximum between the slow-mobile and immobile peaks (Fig. 5b). Compared with the motional states, the fluorescence intensity returned to its initial level within 5 min Cluster size analyses at each time point revealed that after the EGF stimulation, dimers and clusters larger than dimers[27] exhibited transient increases while monomers exhibited transient decreases (Fig. 5c, see the Methods section for cluster size analysis). The membrane residence time of Grb2 was defined as the average duration of trajectories. The temporal changes in the larger clusters were similar to those in the residence time of Grb2 on the plasma membrane, suggesting that large EGFR clusters extended the EGFR-Grb2 interaction upon EGF stimulation (Fig. 5d). The MSD of Grb2 at $\Delta t = 66$ ms also exhibited transient decreases that tended to form large clusters (Supplementary Figure 8c). Overall, upon EGF stimulation, the EGFR gradually underwent a transition from the fast-mobile state to the slow and immobile states within a 270 nm diameter area, which simultaneously formed dimers and large oligomers that may function as signaling hubs for Grb2. Then, the oligomers were decomposed into smaller oligomers (Fig. 5e).

## Discussion

The automated single-molecule imaging system AiSIS successfully acquired clear single-molecule images in all observation fields and avoided the substantial problem of differences in the refractive index between water and cells observed with

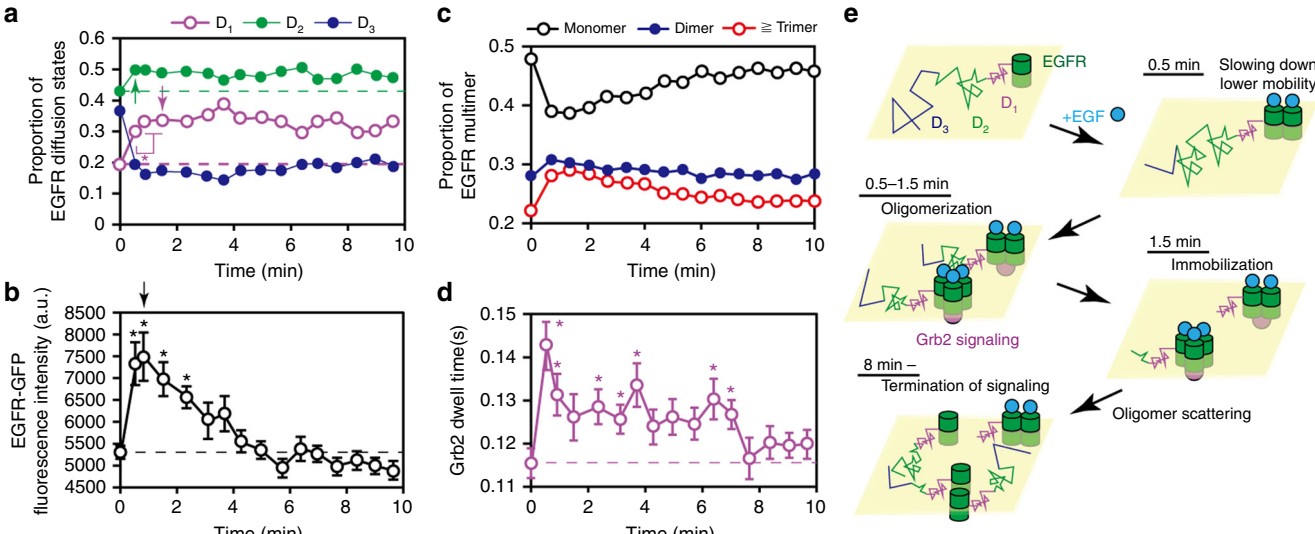

**Fig. 5** Time course of EGFR behavior after EGF stimulation. **a** Temporal changes in the fractions of the three EGFR states. Magenta, green, and blue circles indicate the immobile, slow-mobile, and fast-mobile fractions, respectively. Arrows indicate the first peaks of the immobile and slow-mobile states. The SE is shown in Supplementary Figure 8b. **b** Fluorescence intensity changes of EGFR-GFP. The arrow indicates the maximum intensity. **c** Cluster size changes calculated based on the fluorescence intensity histogram at each time point. Black, blue, and red circles indicate monomers, dimers, and oligomers larger than dimers, respectively. **d** Changes in the residence time of Grb2 on the plasma membrane. The dotted lines shown in **a**, **b**, and **d** indicate data obtained before the EGF addition. Asterisks indicate significant differences compared to the data from before the EGF addition (*$P < 0.05$). The number of data points is shown in Supplementary Table 1. Error bars: SE. **e** Scheme of the EGF-induced (cyan circle) behavioral transitions in EGFR (green cylinders), along with the receptor-evoked cell signaling via Grb2 (magenta circle). Magenta, green, and blue lines indicate immobile, slow-mobile, and fast-mobile EGFR, respectively

conventional focus-keeping systems. Because the autofocusing method utilizes the bright field image of the iris rather than the cells, the algorithm can be applied to all cell types or fluorescent dyes (see the Methods section for details). Auto-cell searching with the deep learning-based technique has two advantages. First, compared with other methods in which the filtering parameters are manually determined by trial and error for each cell line, our method is user friendly, even for users who are not experts in image processing and filtering. Users only need to indicate the area in the image they aim to observe by painting the area using typical computer drawing software, and the input is then used as the training data for subsequent machine learning. Second, compared with manual operation, the photobleaching of fluorescent probes can be minimized using the automated technique because the observed cells are determined from a single snapshot of the fluorescent image taken within 33 ms. These automated technologies for focusing and cell searching are applicable not only to single-molecule microscopy but also to normal optical microscopy at high magnification.

The high statistical precision achieved with sufficiently large amounts of data is required for accurate quantification. In the cases of MSDs, diffusion coefficients, and fluorescence intensity, the data distribution profiles were improved by increasing the number of molecules analyzed (Supplementary Figure 5). The statistical precision was also influenced by cellular heterogeneity, which caused parameter variation (Fig. 2e). Such heterogeneity might lead to an incorrect conclusion when analyzing only a small number of cells. Therefore, large-scale measurements provide more accurate statistical information and facilitate the construction of an appropriate model capable of accurately describing the observed phenomenon.

Compared with conventional measurement methods, AiSIS can perform large-scale single-molecule imaging and analysis more rapidly and easily. When investigating the complicated effects of agonistic and antagonistic drugs, multidimensional measurements should be performed as shown in Fig. 3b and c;

however, numerous samples and conditions are required. Compared with manual operations, the automated AiSIS is well suited to perform these heavy workloads in terms of its precision, reproducibility, and computation time. Furthermore, AiSIS acquires spatiotemporal information from a large number of individual molecules (Fig. 5), thus enabling insights into the dynamic link between molecular behavior and cell signaling.

The developed automation techniques introduced in the present study can process large amounts of data and thus can be used in drug screening, genome-wide screening, and other applications. Although we show the application of these techniques in combination with TIRFM to phenomena in the plasma membrane, our automatic single-molecule imaging analysis can be used for analyses of the cell nucleus and organelles by introducing other microscope techniques, such as oblique illumination, confocal, correlation, and super-resolution microscopy, because the associated systems developed for oil supply, autofocusing, and cell searching are general. Thus, we believe that computer-operated bioimaging using automation and artificial intelligence technologies can lead to a paradigm shift in biological research.

## Methods

**Gene construction**. The cDNA of human EGFR (pNeoSRαII) was provided by Akihiko Yoshimura (Keio University, Tokyo, Japan) and cloned into the pEGFP-C1 vector (Clontech, USA). The linker sequence was the same as that used by Carter and Sorkin[5]. The Grb2-HaloTag was constructed by inserting the Grb2 fragment with BamHI and SalI sites into a Halo7-C2 vector, which was obtained by changing the EGFP sequence in the pEGFP-C2 vector (Clontech, USA) to Halo7 derived from the FN19K HaloTag T7 SP6 Flexi Vector (Promega, USA).

**Cell culture**. CHO-K1 cells were provided by the RIKEN BioResource Center (RIKEN BRC, Japan) and confirmed to be noncontaminated by mycoplasma. Cell lines expressing EGFR-GFP or co-expressing EGFR-GFP and the Grb2-HaloTag were established and used in the experiments. The cells were maintained in HAM F12 medium with 10% fetal bovine serum (FBS) at 37 °C under 5% $CO_2$ and starved in modified Eagle's medium (MEM) without FBS and phenol red for 1 day prior to observation.

**Microscopy**. For single-molecule imaging, the total internal reflection illumination was configured using a high-magnification objective, i.e., PlanApo 60X NA 1.49 (Nikon, Japan), under an inverted microscope (Ti, Nikon, Japan). The Z position of the objective was maintained at a constant position using the pre-equipped objective positioning system Perfect Focus System (PFS, Nikon, Japan). Lasers at wavelengths of 488 nm (Sapphire 488, Coherent, USA) and 561 nm (Sapphire 561, Coherent, USA) were used for excitation of the fluorescent proteins and dyes. The optical filter sets of dichroic mirror/emission filters were DM495/BA500-545 (Nikon, Japan) for the green dyes and DM495/BA500-545 (Nikon, Japan) for the red dyes. The images were acquired at a frame rate of 33 ms using an EMCCD camera (C9100-13, Hamamatsu, Japan). For simultaneous dual-color imaging, an FF493/574 dichroic mirror (Semrock, USA) was installed in the microscope, and a T565lpxr dichroic mirror (Chroma, USA) with emission filters of ET525/50m for green fluorescence and ET605/70m (Chroma, USA) for red fluorescence was set in a multi-image module (Nikon, Japan) connected to two EMCCD cameras. To merge the green and red channels, scattered light images of 60 nm gold particles were acquired in the two channels, and the particle positions were used as the standards for the affine transformation (translation, rotation, and scaling of images in one channel) to compensate for the aberration of the two images.

**Live-cell imaging**. Before each experiment, the culture medium was changed to an imaging solution consisting of Dulbecco's modified Eagle's medium (DMEM) containing 5 mM PIPES, 2% BSA, and 2 mM L-glutamine. The cells were cultured and observed in 60 wells of a 96-well plate, excluding the peripheral wells to avoid interference between the microscope stage and the objective lens adaptor of the oil feeding system (Supplementary Figure 9). A seal was affixed onto the well plate (Rapid Slit Seal, Bio Chromato, Japan) to prevent evaporation of the solution. The observation was performed sequentially along the arrow shown in Supplementary Figure 10a. The solution was mixed with 60 nM EGF to stimulate the cells or used alone as a mock solution. In both cases, the solution was automatically sucked from the well plate using a nozzle connected to a robot arm and dispensed into the target well of the cell culture plate. For the simultaneous imaging of EGFR-GFP and Grb2-HaloTag7, the cells were incubated in 100 pM tetramethylrhodamine HaloTag ligand (Promega, USA) for 15 min.

**Oil feeding system**. Well plate-based measurements are indispensable for large-scale investigations. If the wells are observed from one side of the plate to the other side of the plate using high-magnification microscopy, low levels of immersion oil between the plate bottom and objective lens should be avoided. We developed a novel system that continuously provides the appropriate volume of immersion oil and enables high-magnification observations with long-distance scanning of a plate in one day. The oil flow is driven by a peristaltic pump introduced into the objective lens adaptor (Supplementary Figure 9) and fills the gap between the plate bottom and the lens top. In the case of a low-viscosity medium (e.g., water), the excess volume passively flows along the gradient surface of the adaptor to the peripheral groove and then exits via the downward outlet, even without a sucking mechanism. However, highly viscous immersion oil is not naturally drained from the gap and overflows outside the adaptor rather than through the outlet. Thus, an additional outlet linked to a sucking pump was added to the adaptor. The flow control was effective in maintaining an adequate volume of immersion oil on the lens and allowed for long-term well-based observations.

**Workflow of AiSIS imaging**. A flowchart of the automated procedure is shown in Supplementary Figure 10b. First, the stage was moved to place the target well immediately above the objective lens. Subsequently, autofocusing was performed, and the observable cells were automatically searched up to the required number (~10 cells). In the case of simultaneous EGFR-GFP and Grb2-TMR imaging, cell searching was executed on the EGFR-GFP channel. Then, after readjusting the focus, automatic recordings of single-molecule images of the selected cells were sequentially performed. To investigate the effect of a drug on molecular behavior, the selected cells were divided into two groups and imaged before or after drug addition. Single-molecule tracking and statistical analysis of the acquired single-molecule images were performed in parallel while observing and recording the following cell.

**Autofocusing algorithm**. The autofocusing algorithm consisted of the following two steps: coarse focusing using deep learning and high-precision focusing using image processing. The deep learning-aided method was less precise but more robust in terms of unexpected noise in the iris image, such as debris or bubbles, which were often observed at positions distant from the coverslip. This method was suitable for long Z scanning with a coarse step (750 μm range with a 2.5 μm step). However, the image processing-mediated method was appropriate for fine scanning around the in-focus position (3.5 μm range with a 32 nm step). For autofocusing, the advantages of these methods were appropriately combined and used initially for coarse focusing and then for high-precision focusing.

A flowchart of the coarse focusing process is shown in Supplementary Figure 11a. In advance, we prepared in- and out-of-focus images of the iris as training data for deep learning, and these images were acquired with 0.1 μm steps of the objective around the in-focus position. Both the in- and out-of-focus images

of the iris were used to train the neural network. The detailed procedure used for machine learning is shown in Supplementary Figure 11b. For this learning, adaptive moment estimation (Adam)[28] was applied using Python's library, Chainer (https://chainer.org, Preferred Networks). Two neural network parameters, the convolution/deconvolution weight and the bias values of each layer, were optimized by the learning method and saved in a binary file with other parameters, such as the numbers/types of layers and activation functions (Supplementary Figure 2). The file was loaded and run by a control software programmed using C++ and CUDA on an AiSIS operation PC equipped with GPU (NVIDIA Quadro 4000) during the experiment. For the experiments, the objective lens scanned a predefined Z range of 750 μm at 2.5 μm steps and the obtained iris images were evaluated using deep learning. The evaluation value was determined based on the similarity of the obtained image to the pre-learning in-focus images and set between 0 (not focused) and 1 (focused). In the case of the training data, images obtained within 4 μm of the in-focus objective position were defined as 1 and the other images were defined as 0. Coarse focusing continued until the evaluation value was greater than 0.5. Supplementary Figure 11c shows various iris images and corresponding evaluation values calculated by the trained neural network.

A flowchart of the high-precision focusing process is shown in Supplementary Figure 12a. The objective position, which could be controlled by PFS, was first set to the beginning of a predetermined PFS range of 2200 (a.u.), which corresponded to approximately 3.5 μm. Then, the position was gradually and discretely changed upward with a PFS step size of 20 (a.u.). The sharpness of the iris image was successively calculated and used to evaluate the in-focus and out-of-focus images as shown in Supplementary Figure 12b. To obtain the evaluation value, the SRIC image of the lower half of the iris was captured (Step 1) and binarized using the Otsu method[29] (Step 2). A region of interest (ROI) of $21 \times 21$ pixels was shifted (Step 3) from the top to where the proportion of the white area of the ROI was 50% or less to detect the iris edge (Step 4). This process (Steps 3 and 4) was repeated in the horizontal direction (Step 5), and a brightness histogram of all ROIs on the iris edge in the SRIC image (Step 6) was obtained with 512 bins between the minimum and maximum brightness. Two sharply separated peaks were typically observed when the iris was in focus. The histogram was bisected based on a threshold calculated using the Otsu method (Step 7), and after both regions were smoothed over 20 bins, the brightness corresponding to the peaks $I_{max1}$ and $I_{max2}$ and the valley $I_{min}$ were determined (Steps 8 and 9). Based on these values, the sharpness $E$ was defined as

$$E = N(I_{max1}) \times N(I_{max2})/N(I_{min})^2 \qquad (1)$$

where $N(I)$ is the value of the histogram at intensity $I$. A value of $E > 1$ represents high sharpness. After the objective scanned the entire PFS range, the objective position with the highest $E$ (PFS$_0$) was determined by fitting the sharpness distribution with a Gaussian function as shown in Fig. 1d (right). Finally, the objective was moved to the position where an offset value was added to PFS$_0$ to compensate for the difference between PFS$_0$ and the visually determined in-focus position. The offset value was predetermined.

**Cell searching and cell region detection using deep-learning**. Supplementary Figure 13a provides a schematic diagram of the learning procedure used for cell searching. Single-molecule images of cells expressing various levels of EGFR-GFP were captured in advance. To prepare the training data for machine learning, cell regions with suitable fluorescent spot densities were manually painted, the pixel intensity was set to 1, and the other region was set to 0 (Supplementary Figure 13b). The layer structure of a neural network composed of three convolutions and three deconvolutions is shown in Supplementary Figure 2b. The layer parameters were optimized by learning using Adam and saved in a binary file. The setting file was loaded onto the microscope control computer during the experiment. If an acquired image was not correctly judged by the neural network, the users can incorporate the image into the existing training data and re-execute the learning process to improve the cell searching function. Similarly, by changing the training data according to the researcher's demand, AiSIS can choose cells that are more suitable to the study's purpose. Supplementary Figure 13c provides a flowchart of the cell searching procedure. The fluorescence images were acquired from 225 ($15 \times 15$) fields of view using X–Y scanning, and the region with suitable expression was determined by the trained neural network. After cell searching was completed, image acquisition was performed according to the descending order of the area sizes. The stage position was set at the centroid of the suitable region. Supplementary Figure 13b shows the results obtained from the trained neural network, and it indicates that cells with appropriate fluorescent spot densities were recognized.

Detection of the cell region was performed using the same deep-learning method and SRIC images. The structures of the neural networks are shown in Supplementary Figure 2c. Typical raw images were used as images for learning, and the images obtained from the trained neural network are shown in Supplementary Figure 13d. The cells that adhered to the surface could be correctly recognized.

**Optimization of the number of layers**. The appropriate number of layers in our neural network (Supplementary Figure 14) should be determined by considering the trade-off between the calculation time and the prediction precision. For

evaluation, the prediction by the learned network was compared with the correct answer provided by the researcher. The pixels in the researcher's selected region were assigned values of 1, whereas other pixels were assigned values of 0, and the network outputs provided real number values between 1 and 0. If the values at the $(i, j)$ pixel $(0 \leqq i, j < 512)$ in the researcher's answer and the network output are described as $d_{i,j}$ and $y_{i,j}$, respectively, then the error is defined as the average residual square (ARS)

$$\text{ARS} = \sqrt{\frac{1}{512^2} \sum_{i=0}^{511} \sum_{j=0}^{511} \left(y_{i,j} - d_{i,j}\right)^2} \tag{2}$$

If the network has not learned, then the ARS has a value >0.5. During learning, the ARS value approaches 0 depending on the difficulty of the tasks. The product of the calculation time and ARS was compared among different network types (Supplementary Table 2, Supplementary Figure 14). The number of layers in the networks with the lowest product value was 2, 3, and 3 for coarse autofocusing, cell searching, and cell region detection, respectively. In the case of autofocusing, the calculation time was on the order of milliseconds, even for three layers, which is much shorter than the frame rate of 33 ms. Therefore, networks with three layers were adopted.

We further assessed the types of activation (or transfer) functions, which represent an additional factor in a neural network that affects precision, by referring to the ARS. In the case of coarse autofocusing, the sigmoid function ($f(x) = 1/(1 + e^{-x})$) exhibited a higher ARS than the rectified linear unit (ReLU) function ($f(x) = x \ (x > 0)$, $0 \ (x < 0)$) for the final output layer. Therefore, compared with the other layers for which the ReLU function was used, the sigmoid function was adopted for the final output layer (Supplementary Figure 2a).

When designing the neural network, the procedure was as follows: first, we started from the neural network with a minimum number of layers and weight; second, we increased these values until the ARS was saturated, indicating that the numbers were suitable; and finally, we tried different activation functions of the final layer to design a neural network with the highest speed and the lowest ARS.

**Amount of training data**. A neural network should learn using an appropriate number of training images to avoid underlearning or overlearning. Therefore, we examined the extent of learning using cross validation. The judgment of new images by the network was compared with that of the images used for training using the ARS as shown in Eq. (2). The convergence of the distance between the ARSs was assessed for different extents of training (Supplementary Figure 15). When learning was executed with an appropriate number of data, the distance became constant. Based on our results, the required number of training images was 400, 40, and 200 for coarse autofocusing, cell searching, and cell region detection, respectively.

We estimated the time required for both manual creation of the training images and learning using a neural network. In the case of coarse autofocusing, ~400 training images were required (Supplementary Figure 15a), and these images could be prepared within 4 min using automatic repetitive scanning of the objective lens around the in-focus position. During this process, 100 images were acquired per scan in 50 s. The learning of the neural network was automatically completed within 2 min Thus, at least 6 min is required for preparation of the training images and their learning. In the cases of cell searching and cell region detection, 40 and 200 training images were necessary (Supplementary Figure 15b and c), and the raw image acquisitions were completed within 30 min Subsequently, the suitable regions for the single-molecule imaging and cell adhesion areas in the obtained images were manually painted. The average time required for this process was approximately 3 min per image, regardless of the individual researcher's skill and the image condition. The learning was completed within 10 and 16 min Thus, at least 160 (=30 + 120 + 10) and 646 (=30 + 600 + 16) min preparation are required for automatic cell searching and cell region detection, respectively. In fact, we completed preparation of the training data and learning procedure within one day. Using the neural network, image processing filters were automatically generated based on the easily prepared training data. In contrast, conventional image processing requires the selection and combination of appropriate methods (e.g., density processing, edge extraction, and binarization) based on trial and error; thus, predicting the time required for preparation is difficult. For a wide range of biologists, the use of artificial intelligence could be more efficient and helpful than conventional methods.

**Cell type-dependent learning**. We examined the influence of different cell types on the learning associated with cell region detection using SRIC images (Supplementary Figure 16). We prepared 150 images of CHO or HeLa cells that were divided into 100 and 50 images for the training and test datasets, respectively. Each cell region in the training images was manually painted (Supplementary Figure 16a and b). The neural network shown in Supplementary Figure 2c was assessed for learning from 100 images of CHO or HeLa cells and a combination of 50 images of CHO cells and 50 images of HeLa cells (CHO+HeLa) (Supplementary Figure 16). The ARS defined in Eq. (2) was obtained for predictions of the test dataset of CHO or HeLa cells by different networks trained on the three datasets (CHO, HeLa, and CHO+HeLa). Although Supplementary Figure 16c shows that the precision of the

prediction was the highest when the cell type of the training images was consistent with that of the test images, the networks trained with mismatched cell types could be practically used in the experiment as shown in Supplementary Figures 16d and e. Because the imaging method was the same regardless of the cell type, the cell images exhibited similar features, indicating that training data do not need to be collected from each cell type.

**Single-molecule tracking algorithm**. We developed a software program[26] to perform single-molecule tracking. Although other single-molecule/particle tracking software was available[30,31], our program is convenient for incorporation into our automated system and can be modified according to the experimental purpose. A flowchart of our single-molecule/particle tracking software is shown in Supplementary Figure 17a. First, the cross-correlation between the obtained image and the following two-dimensional Gaussian distribution was calculated.

$$I_{i,j} = \frac{1}{\sqrt{2\pi}\sigma} \exp\left(-\frac{i^2 + j^2}{2\sigma^2}\right) \tag{3}$$

where $(i, j)$ indicates the X–Y position in an ROI (Supplementary Figure 17a, right upper panel) and $\sigma$ indicates the Gaussian standard deviation. The variables $i$ and $j$ were assigned values from −5 to 5 (pixel), and $\sigma$ was set to 2 pixels that cover the entire single-molecule spot. The cross-correlation at $(i, j)$, $y_{ij}$ is described as follows:

$$y_{i,j} = \frac{\sum_{I=-R}^{R} \sum_{J=-R}^{R} \left(I_{I,J} - I_{\text{ave}}\right)\left(x_{i+I,j+J} - x_{\text{ave}}\right)}{\sqrt{\sum_{I=-R}^{R} \sum_{J=-R}^{R} \left(I_{I,J} - I_{\text{ave}}\right)^2} \sqrt{\sum_{I=-R}^{R} \sum_{J=-R}^{R} \left(x_{i+I,j+J} - x_{\text{ave}}\right)^2}}.$$

Here,

$$I_{\text{ave}} = \frac{1}{(2R+1)^2} \sum_{I=-R}^{R} \sum_{J=-R}^{R} I_{I,J},$$

$$x_{\text{ave}} = \frac{1}{(2R+1)^2} \sum_{I=-R}^{R} \sum_{J=-R}^{R} x_{i+I,j+J}, \tag{4}$$

where $x_{i,j}$ and $y_{i,j}$ are the pixel intensities at $(i, j)$ in the obtained and cross-correlated images, respectively, and R is the length of a side of the square ROI (5 pixels). Then, binarization of $y_{i,j}$ using a threshold value of 0.25 and labeling were performed. This threshold should be adjusted depending on the signal-to-noise ratio of the single-molecule image (see also Supplementary Figure 17b). We carefully set this threshold by comparing the binarized image with the raw image by visual inspection to avoid missing small fluorescence signals (threshold ~1) or overcount noise (~0) (Supplementary Figure 17b). Once determined, the threshold value could be applied to another sample/cell because the quality of the images obtained by AiSIS was generally constant due to selection of the imaging field by the trained neural network. Once the centroid of the labeled object was found, fitting within the ROI was executed according to the following function:

$$I\left(x, y; I_0, x_g, y_g, \sigma_A, a, b, I_{\text{back}}\right) = I_0 \exp\left[-\frac{(x - x_g)^2 + (y - y_g)^2}{2\sigma_A^2}\right] + a\left(x - x_g\right) + b\left(y - y_g\right) + I_{\text{back}} \tag{5}$$

where $x$ and $y$ denote the pixel positions. The intensity distribution in the ROI is expressed as a Gaussian function with a peak intensity of $I_0$ at the centroid, $(x_g, y_g)$ and a variance of $\sigma_A^2$ plus a background, which has an inclination described by $a$ and $b$ above the offset intensity $I_{\text{back}}$. The fitted parameters are $I_0$, $x_g$, $y_g$, $\sigma_A^2$, $a$, and $I_{\text{back}}$. To generate a single-molecule trajectory by connecting the spots, the following algorithm is used. All possible connections between two spots at times $t$ and $t − 1$ with a center-to-center distance below 6 pixels are listed. Then, the shortest connection between both spots is selected. Trajectories outside the cell region or with inadequate spot sizes ($\sigma_A < 1.5$ or $\sigma_A > 2.5$) were removed.

**Estimation of EGFR motional states**. The diffusion coefficient $D$ in Fig. 2 was calculated from the step displacement of single-molecule $dr$ during time $T$ (=66 ms) according to the equation: $D = dr^2/4T$. The histogram in Fig. 2c shows the average and standard error of the diffusion coefficient distributions among the cells.

The MSD was calculated from all trajectories in every cell using the following equation:

$$\text{MSD}(n\delta t) = \left\{[x_i(n\delta t + m\delta t) - x_i(m\delta t)]^2 + [y_i(n\delta t + m\delta t) - y_i(m\delta t)]^2\right\}_{i,m}. \tag{6}$$

where $n$ and $m$ are frame numbers, $x_i$ and $y_i$ show the single-molecule position in the $i$-th track, $\delta t$ is the time interval between frames (33 ms), and $[\ ]_{i,m}$ represents the average over $i$ tracks and $m$ frames. The confined length was calculated by fitting the MSD with the following formula[32]:

$$\text{MSD}(t) = \left(C^2/3\right)\left(1 - exp\left(-12Dt/C^2\right)\right), \tag{7}$$

where $C$ and $D$ are the confined length and diffusion coefficient, respectively. We calculated these parameters by the maximum-likelihood method. The deviation between MSD values at time $t$ obtained from the observed cells and the fit seemed to obey normal or log-normal distributions. Although the normal distribution, which is typically used as an error distribution, well described these cell-to-cell variations, the log-normal distribution was more likely to be applied as indicated by the log likelihood ($\log L$) as shown in Supplementary Figure 6a. When the MSD curves was fitted using a maximum likelihood estimation assuming either error distribution, the parameters obtained with the log-normal distribution showed a slightly higher likelihood (Supplementary Table 3) and were almost the same as those obtained with the normal distribution. Therefore, we compared the likelihood of the two distributions. The normal distribution $L_{norm}$ and the log-normal distribution $L_{log}$ are described as follows:

$$L_{norm} = \prod_{i=1}^{N_{cell}} \prod_{j=1}^{15} \frac{1}{\sqrt{2\pi}\sigma_j} \exp\left(-\frac{\left(d_{i,j}-MSD(j/30)\right)^2}{2\sigma_j^2}\right),$$

$$L_{log} = \prod_{i=1}^{N_{cell}} \prod_{j=1}^{15} \frac{1}{\sqrt{2\pi}\sigma_j d_{i,j}} \exp\left(-\frac{\left(\log d_{i,j}-\log MSD(j/30)+\sigma_j^2/2\right)^2}{2\sigma_j^2}\right). \quad (8)$$

where $d_{i,j}$ is the average of the measured MSD in the $i$-th cell at time $j/30$ (s); $N_{cell}$ is the number of cells; and $\sigma_j^2$ represents the variance of the normal or log-normal function for $L_{norm}$ and $L_{log}$, respectively. The log likelihoods are expressed as follows:

$$\log L_{norm} = -\frac{15 N_{cell} \log(2\pi)}{2}$$
$$-\sum_{i=1}^{N_{cell}}\sum_{j=1}^{15}\left\{\log\sigma_j + \frac{\left(d_{i,j}-MSD(j/30)\right)^2}{2\sigma_j^2}\right\},$$

$$\log L_{log} = -\frac{15 N_{cell} \log(2\pi)}{2}$$
$$-\sum_{i=1}^{N_{cell}}\sum_{j=1}^{15}\left\{\log\sigma_j d_{i,j} + \frac{\left(\log d_{i,j}-\log MSD(j/30)+\sigma_j^2/2\right)^2}{2\sigma_j^2}\right\}. \quad (9)$$

After the log likelihood values were maximized, the fitted parameters were $D$, $L$, and $\sigma_j$ ($j = 1, 2, \ldots, 15$). The results are shown in Supplementary Table 3. Since $\log L_{log}$ was larger than $\log L_{norm}$, the parameters in $\log L_{log}$ were adopted.

The probability distribution of the displacement was obtained for each time interval and obeyed the following equation:

$$P(r, \delta t) = \sum_{n=1}^{N} \frac{C_n r}{2 D_n \delta t} \exp\left(-\frac{r^2}{4 D_n \delta t}\right),$$
$$\sum_{n=1}^{N} C_n = 1, \quad (10)$$

where $N$ is the number of states; $C_n$ and $D_n$ are the fraction and the diffusion constant of state $n$, respectively; and $r$ is the displacement. The function of the log likelihood is $\log P(r|\theta)$, where $\theta$ is the parameters $C_n$ and $D_n$ for each state $n$ in the $N$-state model ($N = 1$ to 4) and obtained by the maximum-likelihood method. To determine $N$, we introduced the AIC[33], which was calculated using the following equation:

$$AIC = -2\sum_{i=1}^{N_{mol}} \log P(r_i|\theta) + 2k. \quad (11)$$

where $N_{mol}$ represents the amount of data (displacements during $\delta t = 66$ ms) in one cell and $k$ represents the number of parameters. The $N$-states model with the smallest AIC was adopted according to previous studies[24,25]. Supplementary Figure 7 shows that EGFR and Grb2 were highly likely to adopt 3 states. For each cell, the AIC and parameters of $C_n$ and $D_n$ were acquired.

**Cluster size estimation**. We applied a model in which the intensity histogram of the fluorescent spots was assumed to be dependent on the following probability distribution as in previous single-molecule studies[2,17]:

$$P(x) = \sum_{n=1}^{N} \frac{C_n}{\sqrt{2\pi}n\sigma} \exp\left[-\frac{(x-n\mu)^2}{2n\sigma^2}\right],$$
$$\sum_{n=1}^{N} C_n = 1. \quad (12)$$

where $x$, $n$, and $C_n$ denote the brightness, cluster size, and fraction of the $n$-mer cluster, respectively. $N$ is the maximum number of clusters and was set to 10. In this model, the intensity distribution of the $n$-mer cluster should be the Gaussian function with a center of $n\mu$ and a variance of $n\sigma^2$. The log likelihood is as follows:

$$\log L = \sum_{j=1}^{N_{cell}} \sum_{i=1}^{N_{mol}} \log P\left(d_{i,j}\right). \quad (13)$$

where $d_{i,j}$ is the fluorescence intensity of the $i$-th molecule in the $j$th cell and $N_{cell}$

and $N_{mol}$ are the number of observed cells and molecules in the first frame, respectively. The fitted parameters of $\mu$, $\sigma$, and $C_n$ were obtained by maximizing the log likelihood. Both $\mu$ and $\sigma$ were set as global parameters for our data obtained under different EGF concentrations because the fluorescence properties of GFP and TMR were not affected. In Fig. 5c, changes in the fractions of monomers, dimers, and larger clusters were observed at every 40 s interval after EGF stimulation.

**$EC_{50}$ of the dose–response curve**. The obtained MSD curve plotted against the EGF concentration was fitted using the following equation to calculate the $EC_{50}$:

$$MSD = MSD_{max} - \frac{MSD_{max} - MSD_{min}}{1 + \left(\frac{EC_{50}}{[L]}\right)^h} \quad (14)$$

where $MSD_{max}$ and $MSD_{min}$ are the MSD values of the upper and lower boundaries, respectively, $[L]$ is the ligand (EGF) concentration, and $h$ is the Hill coefficient. To obtain the parameters, we calculated the log likelihood. The deviation between the distribution of MSD values from individual cells and the fit was assumed to depend on the normal or log-normal distribution. The log likelihoods of these distributions are as follows:

$$\log L_{norm} = -\frac{N_{cell} \log(2\pi)}{2}$$
$$-\sum_{i=1}^{N_{cell}}\left(\log\sigma([L]_i) + \frac{\left(d_i-MSD([L]_i)\right)^2}{2\sigma([L]_i)^2}\right),$$

$$\log L_{log} = -\frac{N_{cell} \log(2\pi)}{2}$$
$$-\sum_{i=1}^{N_{cell}}\left\{\log\left(\sigma([L]_i)d_i\right)\right.$$
$$\left. + \frac{\left(\log d_i-\log MSD([L]_i)+\frac{\sigma([L]_i)^2}{2}\right)^2}{2\sigma([L]_i)^2}\right\}, \quad (15)$$

where $\sigma([L])$ indicates the parameters in the normal or log-normal distribution for a condition of $[L]$ nM EGF stimulation. The fitted parameters were $EC_{50}$, $h$, $\sigma([L])$ for $[L] = 0.06, \ldots, 60$ nM. The obtained parameter values are shown in Supplementary Table 4. Because $\log L_{log}$ was larger than $\log L_{norm}$, we adopted the parameters of the log-normal distribution.

When noncompetitive inhibition by AG1478 occurred, the MSD against both the ligand and inhibitor concentrations was assumed to obey the following equation to yield the $EC_{50}$ and $IC_{50}$:

$$MSD = MSD_{max} - \frac{MSD_{max} - MSD_{min}}{\left(1 + \frac{EC_{50}}{[L]}\right)\left(1 + \frac{[I]}{IC_{50}}\right)} \quad (16)$$

where $[I]$ is the inhibitor (AG1478) concentration. Because the error in the data was assumed to obey the normal or log-normal distribution, the parameter was obtained by maximizing the following log likelihoods:

$$\log L_{norm} = -\frac{N_{cell} \log(2\pi)}{2}$$
$$-\sum_{i=1}^{N_{cell}}\left(\log\sigma([L]_i,[I]_i) + \frac{\left(d_i-MSD([L]_i,[I]_i)\right)^2}{2\sigma([L]_i,[I]_i)^2}\right),$$

$$\log L_{log} = -\frac{N_{cell} \log(2\pi)}{2}$$
$$-\sum_{i=1}^{N_{cell}}\left\{\log\left(\sigma([L]_i,[I]_i)d_i\right)\right.$$
$$\left. + \frac{\left(\log d_i-\log MSD([L]_i,[I]_i)+\sigma([L]_i,[I]_i)^2/2\right)^2}{2\sigma([L]_i,[I]_i)^2}\right\}. \quad (17)$$

where $\sigma([L],[I])$ shows the parameters in a normal or log-normal distribution for cells stimulated by $[L]$ nM EGF and $[I]$ nM inhibitors. $[L]_i$, $[I]_i$, and $d_i$ represent the concentrations of the ligand and inhibitor and the average of the measured MSD obtained from the $i$-th cell, respectively; and $N_{cell}$ is the number of cells. The fitted parameters were $EC_{50}$, $IC_{50}$, $\sigma([L, I])$ for $[L] = 0.3, 0.6, \ldots, 60$ nM and $[I] = 0.1, 1, \ldots, 10,000$ nM. The obtained parameter values are shown in Supplementary Table 5. Because $\log L_{log}$ was larger than $\log L_{norm}$, we adopted the parameters of the log-normal distribution.

**Curve fitting**. The maximum-likelihood estimation (MLE) method was used to fit the obtained data to the model equations. The Limited-Memory Broyden Fletcher Goldfarb Shanno algorithm[34] was employed to calculate the diffusion displacement by Eq. (10). A generalized reduced gradient nonlinear optimization method[35] was used to obtain the MSD by Eqs. (7) and (8), the intensity histogram by Eqs. (12) and (13), the $EC_{50}/h$ by Eqs. (14) and (15), and the $EC_{50}/IC_{50}$ by Eqs. (16) and (17). For the fittings, the commercially available software Microsoft Excel (Microsoft) was used. The molecular lateral diffusion was analyzed using Eq. (10) with an in-house-developed program written in Python (https://www.python.org). The least squares method along with the Levenberg Marquardt method[36] were used to fit the

evaluation value distribution for the high-precision autofocusing by the Gaussian function and the intensity profile of the fluorescent spots by Eq. (5).

**Code availability**. All codes are available upon request from the first authors or corresponding authors.

**Data availability**. The data supporting the findings of this study are available from the corresponding authors upon reasonable request.

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

## Acknowledgements

We thank A. Kanayama for providing experimental support, A. Yoshimura for providing the cDNA, M. Takahashi for supporting the molecular cloning, T. Tsuzuki for programing the cell recognition software during the early development stages, and P. Karagiannis for editing the manuscript. This study was supported by the JST-SENTAN program since October 2013 and by AMED-SENTAN since April 2015. This study was also partially supported by AMED-CREST JP17gm0910001.

## Author contributions

M.Y., M.H., J.K., Y.S., and M.U. planned the study; M.Y., M.H., and J.K. constructed the apparatus; M.Y. and J.K. created the automation algorithms; M.H., Y.S., and M.U. designed the experiments; M.Y. and M.H. performed the experiments and analyzed the data; and M.H., M.Y., Y.S., and M.U. wrote the paper.

## Additional information

**Competing interests:** The authors declare no competing interests.

