## [Peer Review File · Nature Communications]

Reviewers' comments:

Reviewer #1 (Remarks to the Author):

The authors claim to have fully automated the single molecule acquisition and analysis process, which could be a very useful step, however it is not clear from the paper whether they have actually developed any novel methods themselves or how the key methods work. While the application of deep learning to some aspects of the process might be a novel application, they provide insufficient explanation how it works to a non-machine learning expert, and no meaningful demonstration that it does work or how widely it could be applied. In contrast to this they provide quite detailed explanation of routine analyses of MSD data. If the novel work is all in cited papers then what is novel here? Some validation would be nice too.

It may be that the idea is that this can be used off-the-shelf by anyone without understanding how it is done?

The way the paper is written gives the reader the impression that the authors are themselves unsure about what they really mean with regard to the deep learning. No detail, so it feels like they used a 'black box'. They often use generic expressions such as "Three convolution layers were designed for the machine learning" (339) (how does the design look like?), "The fitting was performed by using Origin software" (399) (what algorithm was used?) or "Parameters of the neural network were adjusted by learning through training data." (236) With regard to the latter statement: of course they used training data to train the network, but how did they do it? The impression is increased by statements such as "To use the learning result in the microscope control software, we saved the weight and bias of the learned layers as a binary file, ..." (336) Sure they trained the network and saved the results, it would be unusual not to save the results.

The design of the convolution network is apparently much different than the network described in ref 7. However there are no details about the network architecture. A graph of the networks as it has been done in ref 7 would be very helpful.

Specific issues.

They mention the software they use, but do not cite it ("Origin" and "chainer"). They need to state the specific algorithms that have been used to train the network and the fit the model (399). Saying we used chainer or Origin is not enough. The "chainer" website states for instance that it provides "various optimizers". Since these software packages seem central to the paper some words putting them into context should be made.

As far as I can tell they give no justification for the specific choice of the network (why 3 convolution layers and not 5?), and there is little validation; how does the automatic procedures perform relative to the manual procedure, is the automatic procedure optimal or could it be improved?

Explanation of the auto-focusing algorithm. A lot of words are used to explain this and yet it did not make clear to me how it works. Why is a two stage process used? What are the advantages over existing methods? Is it necessary to use deep learning for this? Does the method have to be retrained for each different cell line/dye....? How much data is needed for the training set? How is it tested?

Explanation of cell searching algorithm. Assumes expert knowledge of the machine learning algorithms. Again, does the method have to be retrained for each different cell line/dye and how much data is needed for the training set? How is it tested?

Estimation of EGFR motional states. Analysis of displacement. Shouldn't $P(r,d)$ be $P(r, dt)$? What

type of fit was performed for $P(r, dt)$? Maximum likelihood? What statistical model was assumed to determine the likelihood? This is also important for the use of the BIC for model selection. An appropriate likelihood must be used. Their BIC assumes uncorrelated Gaussian noise - is this appropriate for $P(r, dt)$ given that each time point in each track contributes to multiple points in $P(r, dt)$?

Cluster size estimation. The explanation of S6 is poor. What does "center value" mean? They claim μ is the center value, but if they mean the peak of the Gaussian, from S6 this is actually at $n*\mu$. Also, again, they say nothing about how the fit was performed or the statistics assumed.

Dose response curve fitting. Again no information about the fitting procedure or statistics is provided.

Figure 1e is uninformative, Supplementary Table 1 is unclear. If the size of the layers is given in the table then it is not clear what they are. To state the number of training images is meaningless if it is not put into context; is the number sufficient? could it be reduced? would the result be better if there is more training data? Why did they use almost 3000 images to train the auto focus procedure but not even a 100 for the cell search?

Minor points.

Summary. Line 11. Bad grammar - missing word. "An automated..."

L332. Units? 750m at 2.5m step?

L343/344 Sentence is grammatically incorrect.

L358 Missing word "various levels of ? were captured"

Reviewer #2 (Remarks to the Author):

This is an important study reporting a novel automated single molecule fluorescence microscope. One key bottleneck of single molecule microscopy experiments is the low throughput of often manually performed imaging experiments. The resulting lack of statistical sampling without extremely time consuming experimentation makes systematic variation of parameters typical for pharmacological questions almost impossible. The authors report some significant advance by automatizing the search for cells suitable for imaging using an artificial intelligence method known as deep learning. Other improvements are an objective adapter to provide continuous immersion oil supply and an autofocusing system. They demonstrated the capabilities of the apparatus in a nice exercise in receptor tyrosine kinases pharmacology.

The major shortcoming of the manuscript is that the key methods are presented in insufficient detail. The sections "Auto-focusing algorithm" and "Deep-learning algorithm for cell searching" don't provide nearly enough detail to roughly understand what was done, and they are clearly insufficient to provide the information necessary to reproduce the work. That is a pity! The parameter settings presented in the Supplementary Table 1 lack any context of how they are used. It would be useful to have the training images available and also the code necessary for the deep learning procedure. Another example of lacking interesting detail is the immersion oil refluxing attachment. It would have been nice to show a drawing illustrating a cross-section of the attachment to see how the oil might flow between the objective and the attachment and how it is drained afterwards. Such details would clearly make the manuscript more appealing to a broader community requiring automated high magnification microscopy experiments.

Reviewer #3 (Remarks to the Author):

The manuscript by Yasui et al presents an automated pipeline for single molecule imaging in cells, potentially greatly reducing the learning gradient for the technology and hence making it more accessible. The domain topic is somewhat removed from my expertise; I was asked to comment on the appropriateness of the machine learning methods. These are not described in sufficient detail to allow a full replication; we are only told that a three layer CNN was used, without reference to the explicit architecture (activation functions, size of the filters, etc). Much more information should be given, perhaps in a separate supplementary file. Of critical importance to evaluate the value of the whole approach is the issue of how much training data is necessary. CNNs and deep learning in general is famously data hungry and notoriously uninterpretable. As a consequence of the first, a potentially large, manually curated data set needs to be produced (which in itself could be extremely time consuming). As a consequence of the second, it is usually not obvious how to modify the architecture/ parameters to adapt to a different task (e.g. a different cell type), with the consequence that potentially a completely new training data set might be needed (with consequent time costs). The authors should be clear on these points. Optionally, they may also want to consider whether a less data-hungry approach (e.g. L1 regularised logistic regression on filter-derived features) might not also offer a viable alternative (I appreciate deep learning is more fashionable)

Reviewers' comments:

Reviewer #1 (Remarks to the Author):

The authors claim to have fully automated the single molecule acquisition and analysis process, which could be a very useful step, however it is not clear from the paper whether they have actually developed any novel methods themselves or how the key methods work. While the application of deep learning to some aspects of the process might be a novel application, they provide insufficient explanation how it works to a non-machine learning expert, and no meaningful demonstration that it does work or how widely it could be applied. In contrast to this they provide quite detailed explanation of routine analyses of MSD data. If the novel work is all in cited papers then what is novel here? Some validation would be nice too.

It may be that the idea is that this can be used off-the-shelf by anyone without understanding how it is done?

The way the paper is written gives the reader the impression that the authors are themselves unsure about what they really mean with regard to the deep learning. No detail, so it feels like they used a 'black box'. They often use generic expressions such as "Three convolution layers were designed for the machine learning" (339) (how does the design look like?), "The fitting was performed by using Origin software" (399) (what algorithm was used?) or "Parameters of the neural network were adjusted by learning through training data." (236) With regard to the latter statement: of course they used training data to train the network, but how did they do it? The impression is increased by statements such as "To use the learning result in the microscope control software, we saved the weight and bias of the learned layers as a binary file, ... " (336) Sure they trained the network and saved the results, it would be unusual not to save the results.

The design of the convolution network is apparently much different than the network described in ref 7. However there are no details about the network architecture. A graph of the networks as it has been done in ref 7 would be very helpful.

Specific issues.

They mention the software they use, but do not cite it ("Origin" and "chainer"). They

need to state the specific algorithms that have been used to train the network and the fit the model (399). Saying we used chainer or Origin is not enough. The "chainer" website states for instance that it provides "various optimizers". Since these software packages seem central to the paper some words putting them into context should be made.

As far as I can tell they give no justification for the specific choice of the network (why 3 convolution layers and not 5?), and there is little validation; how does the automatic procedures perform relative to the manual procedure, is the automatic procedure optimal or could it be improved?

Explanation of the auto-focusing algorithm. A lot of words are used to explain this and yet it did not make clear to me how it works. Why is a two stage process used? What are the advantages over existing methods? Is it necessary to use deep learning for this? Does the method have to be retrained for each different cell line/dye....? How much data is needed for the training set? How is it tested?

Explanation of cell searching algorithm. Assumes expert knowledge of the machine learning algorithms. Again, does the method have to be retrained for each different cell line/dye and how much data is needed for the training set? How is it tested?

Estimation of EGFR motional states. Analysis of displacement. Shouldn't $P(r,d)$ be $P(r, dt)$? What type of fit was performed for $P(r,dt)$? Maximum likelihood? What statistical model was assumed to determine the likelihood? This is also important for the of the BIC for model selection. An appropriate likelihood must be used. Their BIC assumes uncorrelated Gaussian noise - is this appropriate for $P(r, dt)$ given that each time point in each track contributes to multiple points in $P(r, dt)$?

Cluster size estimation. The explanation of S6 is poor. What does "center value" mean? They claim μ is the center value, but if they mean the peak of the Gaussian, from S6 this is actually at $n*\mu$. Also, again, they say nothing about how the fit was performed or the statistics assumed.

Dose response curve fitting. Again no information about the fitting procedure or statistics is provided.

Figure 1e is uninformative, Supplementary Table 1 is unclear. If the size of the layers is

given in the table than it is not clear what they are. To state the number of training images is meaningless if it is not put into context; is the number sufficient? could it be reduced? would the result be better if there is more training data? Why did they use almost 3000 images to train the auto focus procedure but no even a 100 for the cell search?

Minor points.

Summary. Line 11. Bad grammar - missing word. "An automated..."

L332. Units? 750m at 2.5m step?

L343/344 Sentence is grammatically incorrect.

L358 Missing word "various levels of ? were captured"

Reviewer #2 (Remarks to the Author):

This is an important study reporting a novel automated single molecule fluorescence microscope. One key bottleneck of single molecule microscopy experiments is the low throughput of often manually performed imaging experiments. The resulting lack of statistical sampling without extremely time consuming experimentation makes systematic variation of parameters typical for pharmacological questions almost impossible. The authors report some significant advance by automatizing the search for cells suitable for imaging using an artificial intelligence method known as deep learning. Other improvements are an objective adapter to provide continuous immersion oil supply and an autofocusing system. They demonstrated the capabilities of the apparatus in a nice exercise in receptor tyrosine kinases pharmacology.

The major shortcoming of the manuscript is that the key methods are presented in insufficient detail. The sections "Auto-focusing algorithm" and "Deep-learning algorithm for cell searching" don't provide nearly enough detail to roughly understand what was done, and they are clearly insufficient to provide the information necessary to reproduce the work. That is a pity! The parameter settings presented in the Supplementary Table 1 lack any context of how they are used. It would be useful to have the training images available and also the code necessary for the deep learning procedure. Another example of lacking interesting detail is the immersion oil refluxing attachment. It would have been nice to show a drawing illustrating a cross-section of the attachment to see how the oil might flow between the objective and the attachment and

how it is drained afterwards. Such details would clearly make the manuscript more appealing to a broader community requiring automated high magnification microscopy experiments.

Reviewer #3 (Remarks to the Author):

The manuscript by Yasui et al presents an automated pipeline for single molecule imaging in cells, potentially greatly reducing the learning gradient for the technology and hence making it more accessible. The domain topic is somewhat removed from my expertise; I was asked to comment on the appropriateness of the machine learning methods. These are not described in sufficient detail to allow a full replication; we are only told that a three layer CNN was used, without reference to the explicit architecture (activation functions, size of the filters, etc). Much more information should be given, perhaps in a separate supplementary file. Of critical importance to evaluate the value of the whole approach is the issue of how much training data is necessary. CNNs and deep learning in general is famously data hungry and notoriously uninterpretable. As a consequence of the first, a potentially large, manually curated data set needs to be produced (which in itself could be extremely time consuming). As a consequence of the second, it is usually not obvious how to modify the architecture/ parameters to adapt to a different task (e.g. a different cell type), with the consequence that potentially a completely new training data set might be needed (with consequent time costs). The authors should be clear on these points. Optionally, they may also want to consider whether a less data-hungry approach (e.g. L1 regularised logistic regression on filter-derived features) might not also offer a viable alternative (I appreciate deep learning is more fashionable)

Response to the reviewers' comments:

We would like to thank the reviewers for their constructive comments. According to the suggestions of all the reviewers, we added considerable details about the methods and algorithms regarding our automated imaging apparatus. Additionally, we described how we determined the number of layers in our neural network and how we prepared the training data for deep-learning. The assessment of the machine learning was also described. Corresponding to reviewer #1, we provided more detail on the fitting and the model evaluation method in the statistical analysis for the experimentally obtained data. According to a suggestion of reviewer #2, we added a cross-section illustration of the objective lens attachment in the immersion oil refluxing system. Finally, we added several Supplementary Figures. Our changes in the manuscript are marked with **Green-colored font**.

List of major changes:

Deep learning for automated single-molecule imaging

Auto-focusing algorithm (page 12~13)

Deep-learning algorithm for cell searching and cell region detection
(page 13~14)

Optimization of number of layers (page 14)

The number of training data (page 14)

Cell type dependent learning (page 15)

Supplementary Figure 2, Figure 9, Figure 10 and Figure 11

Supplementary Table 2

Statistical analysis of single-molecule tracking data

Single molecule tracking algorithm (page 15~16)

Estimation of EGFR motional states (page 16~17)

Supplementary Figure 6

Objective lens attachment

Supplementary Figure 8

Our response to Reviewer #1:

We are grateful to reviewer #1 for the critical comments and useful suggestions that have helped improve our manuscript. As indicated in the responses that follow, we have taken all the comments and suggestions into account in the revised version of our manuscript. We put the comments made by the reviewer in italics with our responses below. Our changes in the revised manuscript are marked with **Green-colored font**.

The authors claim to have fully automated the single molecule acquisition and analysis process, which could be a very useful step, however it is not clear from the paper whether they have actually developed any novel methods themselves or how the key methods work. While the application of deep learning to some aspects of the process might be a novel application, they provide insufficient explanation how it works to a non-machine learning expert, and no meaningful demonstration that it does work or how widely it could be applied. In contrast to this they provide quite detailed explanation of routine analyses of MSD data. If the novel work is all in cited papers then what is novel here? Some validation would be nice too.

It may be that the idea is that this can be used off-the-shelf by anyone without understanding how is it done?

The way the paper is written gives the reader the impression that the authors are themselves unsure about what they really mean with regard to the deep learning. No detail, so it feels like they used a 'black box'. They often use generic expressions such as "Three convolution layers were designed for the machine learning" (339) (how does the design look like?), "The fitting was performed by using Origin software" (399) (what algorithm was used?) or "Parameters of the neural network were adjusted by learning through training data." (236) With regard to the latter statement: of course they used training data to train the network, but how did they do it? The impression is increased by statements such as "To use the learning result in the microscope control software, we saved the weight and bias of the learned layers as a binary file, ... " (336) Sure they trained the network and saved the results, it would be unusual not to save the results.

We understand that the novelty of our method has not been expressed properly in the original manuscript. We added more details about the deep learning process as follows.

Concerning the design of the “three convolution layers for the machine learning”, we have added new **Supplementary Figure 2** in the revised manuscript as a summary. We further describe how the optimal number of the layers was determined in **Methods (page 14)**. In brief, we considered trade-off between the calculation time and the prediction precision. The difference between the correct answer given by the researchers and the prediction by the learned network was calculated and defined as the error. The product of the calculation time and the error was compared between different network types (**Supplementary Figure 9**). The assessment result is shown in **Supplementary Table 2**. The layer numbers with the lowest product values were 2, 3, and 3 for auto-focus, cell searching, and cell region detection, respectively. Based on this finding, we decided to use the networks with three layers. Please see “**Optimization of number of layers**” (**page 14**) of **Methods**.

Concerning “The fitting by using Origin software”, we explained the fitting algorithm was based on the Levenberg-Marquardt method equipped in the non-linear least square fitting tool of Origin software (OriginLab) in “**Cluster size estimation**” (**page 17, line 386-394**) of **Methods**.

Concerning the adjustment of the “parameters of the neural network”, we showed the parameters in new **Supplementary Figure 2**. We explained optimization of the network including the use of adaptive moment estimation (Adam) in “**Auto-focusing algorithm**” (**page 12~13**) and “**Deep-learning algorithm for cell searching and cell region detection**” (**page 13~14**) of **Methods**. Also we described the appropriate number of training data for the machine learning in “**The number of training data**” (**page 14**) of **Methods** and **Supplementary Figure 10**.

In regards to “*The impression is increased by statements such as "To use the learning result in the microscope control software, we saved the weight and bias of the learned layers as a binary file, ... " (336) ..., it would be unusual not to save the results.*”, we have changed the expression “learning result” to “learned neural network” on **line 279 of page 12** for clarity. The information of the learned neural network should be saved in a binary file for transfer between different PCs during the learning and microscope control. The information saved in the binary file is indicated on **page 12~13, line 283-285**.

The design of the convolution network is apparently much different than the network described in ref 7. However there are no details about the network architecture. A

graph of the networks as it has been done in ref 7 would be very helpful.

We have added illustrations of the network architecture to **Supplementary Figure 2** and **Supplementary Figure 9**.

Specific issues.

They mention the software they use, but do not cite it ("Origin" and "chainer"). They need to state the specific algorithms that have been used to train the network and the fit the model (399). Saying we used chainer or Origin is not enough. The "chainer" website states for instance that it provides "various optimizers". Since these software packages seem central to the paper some words putting them into context should be made.

We cited the names of software at appropriate positions in the revised manuscript (**page 12, line 277-278; page 17, line 393-394**). In addition, details of the optimizer in our network were described in **Methods (page 12, line 277; page 13, line 308)**.

As far as I can tell they give no justification for the specific choice of the network (why 3 convolution layers and not 5?), and there is little validation; how does the automatic procedures perform relative to the manual procedure, is the automatic procedure optimal or could it be improved?

The neural network structures in our system (**Supplementary Figure 2**) were determined by considering trade-off between the calculation time and the precision of the results as described in our reply to general comments. The product of the calculation time and the error was compared between different network types. As shown in revised **Supplementary Table 2**, the numbers of layers in the network with the lowest product value were 2, 3, and 3 for auto-focus, cell search, and cell region detection, respectively. Therefore, the networks with three layers were adopted for the functions. The calculation time for the cell search was less than 1 second, sufficiently shorter for high throughput single-molecule imaging. Our apparatus obtained clear single-molecule images and performed single-molecule tracking combined with statistical analysis for 1,600 cells in one day. This processing speed has never been accomplished by manual operators. We did not include a comparison between throughputs by the apparatus and

by researchers in the manuscript to avoid ambiguity about the throughput estimation of the manual procedure. Our achievement will be a measure for further development in high throughput automated single-molecule imaging.

Explanation of the auto-focusing algorithm. A lot of words are used to explain this and yet it did not make clear to me how it works. Why is a two stage process used? What are the advantages over existing methods? Is it necessary to use deep learning for this? Does the method have to be retrained for each different cell line/dye....? How much data is needed for the training set? How is it tested?

The first focusing process moves the objective lens to where the image of the field stop can be recognized. While the objective is approaching the coverslip, deep learning is used to judge whether the image is clear enough to perform the second focusing process (**Fig. 1a**). The second auto-focusing process utilizes the pixel intensity distribution around the edge region of the field stop image to achieve highly precise focusing, in which deep learning is not helpful. Even when the field stop image is blurred or not observed due to the distant initial objective position, our two-step procedure works well. We described this explanation in “**Auto-focusing algorithm**” **Methods (page 12, line 271-)**.

The algorithm for auto-focusing can be applied regardless of the cell type or dye because the method of auto-focusing utilizes the bright field image of field stop but not cells. However, for automatic cell region detection, cell type-dependent machine learning is required because of the cell type-specific appearance, as described below. Accordingly, we added “**Cell type dependent learning**” to **Methods (page 15, line 335-)**. The number of data needed for the machine learning was determined based on the extent of overlearning by cross validation in which the obtained images of a specific cell type(s) were used for both the test and training data sets. We found approximately 1,000 teacher data sets were enough for the training (**Supplementary Figure 10**). We added this information to “**The number of training data**” in **Methods (page 14, line 327-)**.

Explanation of cell searching algorithm. Assumes expert knowledge of the machine learning algorithms. Again, does the method have to be retrained for each different cell line/dye and how much data is needed for the training set? How is it tested?

The search for cells suitable for single-molecule imaging was carried out by taking snapshots of the fluorescence images. Because the appropriate density of single-molecules is almost the same regardless of the cell type or fluorescent protein/dye, training for each different condition is not necessary. Actually, we could choose suitable cells based on the same training data for different fluorescent protein/dyes (GFP and tetra-methyl rhodamine). On the other hand, automatic cell region detection could be sensitive to the cell-type specific appearance obtained by SRIC imaging. Therefore, we checked the validity of learning for different cell lines of CHO and HeLa cells by calculating the differences between the prediction and the correct answer. The results indicated the necessity of training for every cell type (**Supplementary Figure 11**). We added this information in “**Cell type dependent learning**” of **Methods (page 15, line 335-)**.

Estimation of EGFR motional states. Analysis of displacement. Shouldn't $P(r,d)$ be $P(r, dt)$? What type of fit was performed for $P(r,dt)$? Maximum likelihood? What statistical model was assumed to determine the likelihood? This is also important for the of the BIC for model selection. An appropriate likelihood must be used. Their BIC assumes uncorrelated Gaussian noise - is this appropriate for $P(r, dt)$ given that each time point in each track contributes to multiple points in $P(r, dt)$?

About the estimation of EGFR motional states, we corrected **equation S6** on **page 16** and reconsidered the fitting and the model selection. In the revised manuscript, we adopt the maximum likelihood method to obtain parameters in $P(r,dt)$ and Akaike information criterion (AIC) to estimate the number of motional states. These methods are more appropriate for the obtained data and have been previously reported (*Biophys. J.* **97** 1115–1124 (2009) and *PLOS Computational Biology* **9** e1002862 (2013)). Calculation of the maximum likelihood was performed using the Limited-memory Broyden Fletcher Goldfarb Shanno algorithm. Based on the results, we modified **Figures 4cd, 5a, Supplementary Figure 6** and **7ab** and changed “**Estimation of EGFR motional states**” in **Methods** on **page 16~17, line 366-385**.

Cluster size estimation. The explanation of S6 is poor. What does "center value" mean? They claim μ is the center value, but if they mean the peak of the Gaussian,

from S6 this is actually at $n \cdot \mu$. Also, again, they say nothing about how the fit was performed or the statistics assumed.

In regards to the comment on cluster size estimation, we rewrote the explanation of original equation S6 (**revised equation S8**) to read, “Gaussian probability function for the intensity distribution of n -mer cluster is defined with a center value of $n\mu$ and a variance of $n\sigma^2$.” (**page 17, line 389-**). The intensity distribution was fitted by the non-linear least square fitting tool of Origin software (OriginLab) using the Levenberg-Marquardt method.

Dose response curve fitting. Again no information about the fitting procedure or statistics is provided.

The non-linear least square fit was performed on the curves in **Figure 3a and c** using the Generalized Reduced Gradient nonlinear optimization method. We also added one sentence in **Methods (page 18, line 404-405)** to explain this.

Figure 1e is uninformative, Supplementary Table 1 is unclear. If the size of the layers is given in the table than it is not clear what they are. To state the number of training images is meaningless if it is not put into context; is the number sufficient? could it be reduced? would the result be better if there is more training data? Why did they use almost 3000 images to train the auto focus procedure but no even a 100 for the cell search?

We removed Supplementary Table 1 in the original manuscript and instead, we placed the values at appropriate positions in revised **Supplementary Figure 2** that represents the neural network structures in our system. We added the procedure to evaluate enough number of training data in “**The number of training data**” of **Methods** and new data in **Supplementary Figure 10**.

We determined the number of training data as follows. Pixels in regions where researchers or the learned network considered iris, suitable density of fluorescence spot, or cells were set to 1, while the values of other pixels were 0. The difference in values of the corresponding pixels between the correct answer given by the researchers and the predicted result given by the neural network was defined as the error. When new images

were presented as the test data, the convergence of the distance between the errors of these conditions was referred to assess the extent of overlearning. For the first auto-focus and the cell search with an appropriate density of single molecules, at least approximately 1,000 and 100 training images were sufficient to reduce the distance to almost zero, respectively. In the case of cell region detection, 100 training images made the distance constant. Based on these results, we determined the number of training data. Accordingly, we added one paragraph in “**The number of training data**” of **Methods** (page 14).

Minor points.

Summary. Line 11. Bad grammar - missing word. "An automated..."

L332. Units? 750m at 2.5m step?

L343/344 Sentence is grammatically incorrect.

L358 Missing word "various levels of ? were captured"

Thank you for pointing out our mistakes. We corrected them as the followings.

In **Summary**, the first sentence was changed to "An automated...".

In **line 274** of the revised manuscript, the values and units are 750 μm for predefined scanning range and 2.5 μm for step displacement.

In **lines 288-289** of the revised manuscript, the sentence was corrected to "... the sharpness of the iris edge was successively calculated from the obtained images and used for the evaluation of being in-focus or out-of-focus."

In **line 304** of the revised manuscript, we add the missing word "EGFR-GFP".

Our response to Reviewer #2:

We are grateful to reviewer #2 for the critical comments and useful suggestions that have helped improve our paper. We provide point-to-point answers to the comments and added detailed explanation in the revised manuscript. We put the comments made by the reviewer in italics with our responses below. Our changes in the manuscript are marked with **Green-colored font**.

This is an important study reporting a novel automated single molecule fluorescence microscope. One key bottleneck of single molecule microscopy experiments is the low throughput of often manually performed imaging experiments. The resulting lack of statistical sampling without extremely time consuming experimentation makes systematic variation of parameters typical for pharmacological questions almost impossible. The authors report some significant advance by automatizing the search for cells suitable for imaging using an artificial intelligence method known as deep learning. Other improvements are an objective adapter to provide continuous immersion oil supply and an autofocus system. They demonstrated the capabilities of the apparatus in a nice exercise in receptor tyrosine kinases pharmacology.

We greatly appreciate this positive evaluation of our work.

The major shortcoming of the manuscript is that the key methods are presented in insufficient detail. The sections “Auto-focusing algorithm” and “Deep-learning algorithm for cell searching” don’t provide nearly enough detail to roughly understand what was done, and they are clearly insufficient to provide the information necessary to reproduce the work. That is a pity! The parameter settings presented in the Supplementary Table 1 lack any context of how they are used. It would be useful to have the training images available and also the code necessary for the deep learning procedure.

In the revised manuscript, we provided detailed descriptions of the algorithms for the auto-focusing and deep learning for cell searching in **Methods (pages 12~14)**. In addition, we described the methods for “**Optimization of number of layers**” and “**The number of training data**” in **Methods (page 14)**. Other details about methods were added as follows in the revised manuscript.

• Auto-focusing algorithm

1. The roles of the sequential procedures in the algorithm were described in detail as deep learning-based Z scanning to find the field stop image in the first coarse focusing and image processing based-precise positioning in the second fine focusing (**page 12~13, line 271-, Auto-focusing algorithm**).

• Deep-learning algorithm

1. The deep learning process consisting of convolution and deconvolution layers was drawn in **Supplementary Figure 2**.
2. To determine the appropriate number of layers in the deep learning, trade-off between the calculation time and the precision of the results was considered. Products of the time for the calculation and the acquired precision were compared between different numbers of layers (**Supplementary Figure 9**), and the results are shown in **Supplementary Table 2**. Based on the results, we decided to use three layers for both the convolution and deconvolution layers. Please see “**Optimization of number of layers**” of **Methods (page 14)**.
3. The neural network was optimized with adaptive moment estimation (Adam). Please see “**Auto-focusing algorithm**” and “**Deep-learning algorithm for cell searching and cell region detection**” of **Methods (page 12~14)**.
4. The number of data required for the learning was determined by assessing the extent of overlearning to different numbers of training data. **Supplementary Figure 10** summarizes the results and shows that learning from 100 training data is sufficient for cell search and cell region detection, while approximately 1,000 data is required for auto-focus. Accordingly, we added one paragraph to “**The number of training data**” of **Methods (page 14)**.

• Other points

1. Single-molecule tracking. An explanation of the tracking algorithm was described in “**Single molecule tracking algorithm**” of **Methods (page 15~16, line 351-)**.

In addition, we removed Supplementary Table 1 in the original manuscript and instead, we placed the values at appropriate positions in revised **Supplementary Figure 2**.

We show a typical example of the training images for “cell region detection” in **Supplementary Fig. 11**, in which the cell regions were painted in red by the researchers. We prepared 100 training images after determining the sufficient number of training data. Please see **Methods (page 13-14)** for details of the machine learning procedure regarding automatic cell recognition and **Supplementary Figure 10** for the results. We did not include the program code in the manuscript because the code can be

produced simply with standard libraries (e.g. Chainer) based on the information in **Supplementary Figure 2**.

Another example of lacking interesting detail is the immersion oil refluxing attachment. It would have been nice to show a drawing illustrating a cross-section of the attachment to see how the oil might flow between the objective and the attachment and how it is drained afterwards. Such details would clearly make the manuscript more appealing to a broader community requiring automated high magnification microscopy experiments.

To show the detail of the immersion oil refluxing attachment, we added an illustration of the cross-section of the attachment and the immersion oil flow in **Supplementary Figure 8** in the revised manuscript.

Our response to Reviewer #3:

We are grateful to reviewer #3 for the critical comments and useful suggestions that have helped improve our paper. As indicated in the responses that follow, we have addressed all the comments and suggestions into the revised version of our paper. We put the comments made by the reviewer in italics with our responses below. Our changes in the manuscript are marked with **Green-colored font**.

The manuscript by Yasui et al presents an automated pipeline for single molecule imaging in cells, potentially greatly reducing the learning gradient for the technology and hence making it more accessible. The domain topic is somewhat removed from my expertise; I was asked to comment on the appropriateness of the machine learning methods. These are not described in sufficient detail to allow a full replication; we are only told that a three layer CNN was used, without reference to the explicit architecture (activation functions, size of the filters, etc). Much more information should be given, perhaps in a separate supplementary file.

We have added **Supplementary Figure 2** to represent our deep learning architecture with the activation functions and the filter size. We additionally wrote details in **Methods** of the revised manuscript that include the following points.

1. We confirmed by considering the trade-off between the calculation time and the precision of the predictions (**Supplementary Table 2**) that the deep learning process in our apparatus was appropriate for three convolution and three deconvolution layers (**Supplementary Figure 2**). The layer numbers with the lowest cost were found to be 2, 3, and 3 for auto-focus, cell searching, and cell region detection, respectively. Based on this finding, we decided to use the networks with three layers. We added two paragraphs to explain the procedure, “**Optimization of number of layers**” and “**The number of training data**”, in **Methods (page 14, line 316-)**.
2. Optimization of the neural network was done with adaptive moment estimation (Adam), as described in “**Auto-focusing algorithm**” and “**Deep-learning algorithm for cell searching and cell region detection**” (page 12~14) of **Methods**.
3. Both auto-focusing and cell searching showed high reproducibility regardless of cell type, therefore, training for every cell line is not required. On the other hand, automatic cell region detection was sensitive to the cell type based on the SRIC appearance, as described below. Please see **page 5 line 93-95** and **Supplementary Figure 11**.

4. The appropriate number of training data was assessed by the extent of overlearning for different numbers of trainings, as described below.

Of critical importance to evaluate the value of the whole approach is the issue of how much training data is necessary. CNNs and deep learning in general is famously data hungry and notoriously uninterpretable. As a consequence of the first, a potentially large, manually curated data set needs to be produced (which in itself could be extremely time consuming). As a consequence of the second, it is usually not obvious how to modify the architecture/ parameters to adapt to a different task (e.g. a different cell type), with the consequence that potentially a completely new training data set might be needed (with consequent time costs). The authors should be clear on these points. Optionally, they may also want to consider whether a less data-hungry approach (e.g. L1 regularised logistic regression on filter-derived features) might not also offer a viable alternative (I appreciate deep learning is more fashionable).

We determined the number of training images as follows. The prediction errors of the deep learning, which was defined as the difference between the values of corresponding pixels in the original and predicted images, were calculated for different numbers of training images. When new images were examined as the test data to assess the extent of overlearning, the distance between their error values approached zero at approximately 1,000 and 100 training images for the auto-focus and cell search, respectively (**Supplementary Figure 10**). The distance reached a constant at 100 training images for the cell region detection. Based on these results, we determined the number of training images. Please see “**The number of training data**” (**page 14, line 327-334**) of **Methods**.

The training data for deep learning in the auto-focus, cell search, and cell region detection are easy to be produced by simply enclosing the suitable region by hand in the field stop image, single-molecule fluorescence image, and bright field (SRIC) image of cells, respectively. Even though the required number of training data reaches 1,000 images, it takes less effort and time to complete these procedures than to create reliable and precise filters for non-machine learning methods in a trial-and-error fashion by an expert on image processing. Therefore, the deep learning based method should be superior to other approaches.

In the case of auto-focus and cell search, suitable regions in the training image are independent of the cell type, therefore, parameter adaptation for each cell type is not

necessary. We added one sentence to describe this point on **page 5, line 93-95**. On the other hand, for the cell region detection, the learning process indicated the necessity of training for every cell type, as shown in **Supplementary Figure 11**. In brief, we used the SRIC images for the automatic cell region detection. We checked the validity of learning for different lines of CHO and HeLa cells by calculating the differences between the prediction and the correct answer. The results show the cell region detection was sensitive to the cell-type specific appearance obtained by SRIC. Please see “**Cell type dependent learning**” (**page 15, line 335-350**) of **Methods** and **Supplementary Figure 11**.

Reviewers' comments:

Reviewer #1 (Remarks to the Author):

The manuscript has clearly improved. The added paragraphs and figures are useful for the understanding of the method. The authors have clearly made an attempt to address the issues, and I believe the work, particularly the application of machine learning to automate the initial stages of the process, is good and should be published. The clarity of the writing has improved; it is still not great but acceptable. However, I would recommend the authors do the important improvements suggested below. This is really important if the authors want to reach out beyond core experts and get properly cited

Some issues remaining:

The URL addresses for the "chainer" and "Origin" websites should be cited in the text according to the guide lines. The "chainer" website gives a citation for the Adam optimiser. However, I recommend to include a direct citation into the manuscript. P. 14 line 345 presumably refers to equation 1. There should be a reference instead of "defined above". I do not understand what is meant by "the number of feature quantities ... was multiplied by 1.5" (p15 line 342). This should be expressed more clearly. References for the Otsu method p.13, the L-BFGS algorithm (if not contained in ref 23 or 24) p.17 and General Reduced Gradient p18. should be included into the manuscript. Even for fairly known methods references should be given, not just to guide readers from a range of different fields, but also because their usage and implementation may vary.

The detailed description of the second part of the autofocussing algorithm from about 294 to 302 is so hard to follow that I still don't think I fully understand, and the Figure 1d and its caption in particular provide no help. They also do not reference the figure at this stage. They use the word "value" so many times in the paragraph which is completely ambiguous. They should use words which make clear the quantity they are using.

Single molecule tracking. There is no discussion of any testing of how well this algorithm works, whether the thresholds are appropriate, how they were chosen, when the parameters of the algorithms will need tuning to different data and problems. How would any such tuning affect the automation? For a generally applicable automated system this discussion is important. Could/should other more sophisticated tracking algorithms be used instead?

The authors have provided a little more information about the statistics used in the various fits in their post-tracking analysis used to illustrate the system, but not enough, and they still do not justify the suitability of any of these statistics, e.g. is least squares suitable where they have used it?

378-385. Estimation of motional states. The authors still do not state and justify which likelihood was used.

389-393. Confusing language. On line 389 "n" is the cluster size. In line 392 the authors appear to say $\sqrt{n} \cdot C_n \cdot \sigma$ is the size.

295: what is the "value" of the peak?

295: do they mean the brightness histogram of the original image in the ROI or of the binarized image? I'm guessing they mean the former.

299: highest "value" of what? Distribution of the "values" of what? I could not work this out. I have no idea what the Gaussian function has been fitted to.

318: "The difference between..." This sentence is confusing. The difference between which

quantity per pixel predicted by the network is compared to the correct value for that quantity per pixel?

404-405: The authors should state in the text that they do a least squares fit as explained in their response.

Examples of confusing language (not exhaustive):

90-93: Gives the impression that a problem with conventional systems is that they successfully acquire clear images. If I correctly understand their intended meaning, then a less confusing possibility is "The method successfully acquires clear single-molecule images for all observation fields, avoiding a substantial problem in conventional focus-keeping systems that comes from a difference in the refractive index between water and cells."

Minor language points I noticed (also not exhaustive):

303: "image in cells" - "images of cells"

307: "designed" - wrong word? Perhaps "chosen"?

308: "with applying Adam" - "using Adam"

311-312: "center of gravity" - "centroid"

Reviewer #2 (Remarks to the Author):

Remaining major issues in the revised manuscript:

1) It is not clear how the error values in the machine learning procedures were defined. Are these incorrectly assigned pixels? Or totally missed targets?

This question could be addressed by providing a training image data set (microscope image plus researcher generated/painted answer) together with deep learning generated answer images.

Let's say the cell in an image covers only 20 percent of all pixels, then how much of the cell area is incorrectly recognized given an error of 0.1? Is that 10 percent of all pixels in the image, or about half of the cell?

The revised manuscript falls short of clarifying the deep learning algorithm as requested by all three referees. The figures sketching the dimensionality reduction are insufficient.

2) The description of the symbols used in Equation 2, the two-dimensional gaussian distribution, makes not much sense. You say that i and j are each between $-ROI$ and $+ROI$. Why do you subtract ROI from i and j in the gaussian? Why do you call $I_{\{i,j\}}$ the cross-correlation value at (i,j) ? Isn't it just the intensity of the gaussian? You said that you calculated the cross-correlation between the image and the two-dimensional gaussian. That's not what the equation shows. Very confusing!

Reviewer #3 (Remarks to the Author):

I thank the authors for their detailed letter. My feeling is that the paper still falls short of a comprehensive explanation of the neural network approach and demonstration of its benefits. Notably, the description of the amount of data needed is very vague in the main text (e.g. the number of training data is mentioned as 1000 in the response letter but not mentioned in the text). I am still unconvinced that a manual annotation of 1000 images is a price worth paying for a method that will need to be redesigned for every new cell type. Could we quantify how much time/effort is saved overall if we take into account preparation of the training set, selection of the model, etc, compared to a standard study? Additionally, I feel the description of some of the choices is still lacking, e.g. how to choose the number of layers is very poorly described, and the choice of activation function/ filters is not justified at all.

Response to the reviewers' comments:

Our response to Reviewer #1:

We thank reviewer #1 for the important suggestions that have helped improve our manuscript. As indicated in the following responses, we have taken all the comments and suggestions into account in the revised version of our manuscript. We put the comments made by the reviewer in *bold italics* with our responses below. Our changes in the revised manuscript are marked with highlighted **Green-colored font**.

The manuscript has clearly improved. The added paragraphs and figures are useful for the understanding of the method. The authors have clearly made an attempt to address the issues, and I believe the work, particularly the application of machine learning to automate the initial stages of the process, is good and should be published. The clarity of the writing has improved; it is still not great but acceptable. However, I would recommend the authors do the important improvements suggested below. This is really important if the authors want to reach out beyond core experts and get properly cited

We thank the reviewer for the positive and supportive comments. To explain our machine learning approach for automated bio-imaging analysis with single-molecule sensitivity to a wide range of biological researchers, we have added detailed descriptions with several new figures in the revised manuscript. We believe that our revised manuscript will make our findings more convincing to users not familiar with this work.

Some issues remaining:

The URL addresses for the "chainer" and "Origin" websites should be cited in the text according to the guide lines. The "chainer" website gives a citation for the Adam optimiser. However, I recommend to include a direct citation into the manuscript. P. 14 line 345 presumably refers to equation 1. There should be a reference instead of "defined above". I do not understand what is meant by "the number of feature quantities ... was multiplied by 1.5" (p15 line 342). This should be expressed more clearly. References for the Otsu method p.13, the L-BFGS algorithm (if not contained in ref 23 or 24) p.17 and General Reduced Gradient p18. should be included into the manuscript. Even for fairly known methods references should be given, not just to guide readers from a range of different fields, but also because their usage and

implementation may vary.

As the reviewer suggests, we cited the URL address of “Chainer” (<https://chainer.org>) and “Origin” (<https://www.originlab.com>) in the main text of the revised manuscript (page 15 lines 334-335 for Chainer and page 26 line 581 for Origin). Following the reviewer’s suggestion about the ambiguous expression “*defined above*” in the original manuscript, we changed both to indicate the correct citation of the corresponding equation (Eq. 2 in the revised manuscript, line 410).

Concerning “*the number of feature quantities ... was multiplied by 1.5*”, we meant that the depth of the convolution layer in Supplementary Figure 13k is 1.5 times larger than that in Supplementary Figure 13i to examine the dependency of the cell type-dependent learning on “the number of feature quantities”. Please see “Cell type-dependent learning” (page 20-21, lines 458-481) of the Methods section and Supplementary Figure 13k.

We cited the reference for the Otsu method in the main text (ref. 29). We also modified “Curve fitting” of the Methods section (page 26 lines 573-585) by adding references for the L-BFGS algorithm, the General Reduced Gradient, and related methods.

The detailed description of the second part of the autofocussing algorithm from about 294 to 302 is so hard to follow that I still don't think I fully understand, and the Figure 1d and its caption in particular provide no help. They also do not reference the figure at this stage. They use the word "value" so many times in the paragraph which is completely ambiguous. They should use words which make clear the quantity they are using.

We thank the reviewer for the comments. To provide a better description of the algorithm, we added flowcharts and schematics of the auto-focusing procedures in Supplementary Figure 10 (coarse autofocusing) and Supplementary Figure 11 (high-precision autofocusing). In the original manuscript, we used the terms “first auto-focusing” and “second auto-focusing”. To represent the roles of these procedures more clearly, we changed these terms to “coarse autofocusing” and “high-precision autofocusing” in the revised manuscript. We also provide more description in “Autofocusing algorithm” of the Methods section on page 15-17 with Supplementary Figures 10 and 11. Further, we reference the figures in this section as suggested by the reviewer.

Concerning the usage of “value(s)”, we have modified our use to avoid ambiguity.

With these changes, we hope that our explanation of the autofocussing procedures has become more convincing to the readers.

Single molecule tracking. There is no discussion of any testing of how well this algorithm works, whether the thresholds are appropriate, how they were chosen, when the parameters of the algorithms will need tuning to different data and problems. How would any such tuning affect the automation? For a generally applicable automated system this discussion is important. Could/should other more sophisticated tracking algorithms be used instead?

We thank the reviewer for the important comments. The molecular behavioral parameters obtained by using our tracking software, which include diffusion coefficients, MSD, and cluster size, coincided well with those observed in previous studies (ref. 17), showing that our algorithm could be used to measure living cells. This agreement is mentioned with the measured parameters and references on page 8 (lines 175-177) in the main text.

As the reviewer points out, parameter settings such as the threshold, ROI size and the variances of Gaussian functions are important for single-molecule/particle tracking. To explain the parameter settings for single-molecule/particle tracking, we added new Supplementary Fig. 16 in the revised manuscript. As shown in Supplementary Fig. 16, based on the size and brightness of the fluorescent spots, we carefully set the threshold and other parameters by visually comparing the binarized images with the raw images. Once these parameter values were fixed, no critical tuning was required to obtain a large amount of reproducible single-molecule data, because the images obtained by AiSIS were not affected by the experimental conditions. This independency from the experimental conditions was because the cell searching process by a neural network mediated the image filtering. In fact, AiSIS constantly chose cells suitable for the acquisition of single-molecule images (Fig. 1b), even though the cells used in our study showed various expression levels of EGFR-GFP. We have added a detail description of the single-molecule/particle tracking algorithm in “Single-molecule tracking algorithm” (page 21-23) of the Methods section and Supplementary Fig. 16.

There is an abundance of single-molecule tracking software provided either commercially or as open source. However, we recommend our program, especially when incorporating AiSIS and our experiment aims, because we can check the source code and operation by ourselves. This point is stated on page 21 (lines 485-487) in the revised manuscript. As the reviewer suggests, we may incorporate other sophisticated

tracking algorithms in the future after carefully evaluating the source code, operation, and compatibility with our automated imaging system. At the present stage of our development, we prefer our program for the reasons stated above.

Reviewer #2, comment #2 is related to this comment. Please also see our response to that comment.

The authors have provided a little more information about the statistics used in the various fits in their post-tracking analysis used to illustrate the system, but not enough, and they still do not justify the suitability of any of these statistics, e.g. is least squares suitable where they have used it?

We thank the reviewer for the important comment. We reconsidered the fitting method for the observed data (e.g. Figs. 2c-d and 3a-c) and decided maximum likelihood estimation (MLE), not least square estimation (LSE), is best. When we assume the distribution of residuals between the observed data and the model is Gaussian, MLE is equivalent to LSE, as described in statistics textbooks. However, MLE has some useful advantages over LSE. For example, MLE uses more information in the entire data set than LSE, and MLE is versatile and applicable to various types of data and models. In the revised manuscript, we described the MLE method on pages 23-26. In brief, the residuals between the data and model equations (Eqs. 7, 12, 14, and 15) were assumed to obey a Gaussian function, and thus the log-likelihoods of the model equations were obtained as Eqs. 9, 13, and 16. For example, Eq. 9 is as follows,

$$-\frac{n}{2}\log(2\pi) - \sum_{i=1}^n \log \sigma_i - \sum_{i=1}^n \frac{(d_i - MSD(t_i))^2}{2\sigma_i^2}, \quad (9)$$

where d_i and σ_i is the average and the standard deviation of the measured MSD at time t_i , and n is the number of data. Eqs. 13 and 16 have similar formulae. We maximized the third term in the fitting, because the first two terms were constant regardless of the parameters. The fitting methods used to maximize the likelihood are described in “Curve fitting” of the Methods section on pages 26 (lines 573-585) of the revised manuscript. We hope these descriptions are sufficient to resolve the concerns raised by the reviewer.

378-385. Estimation of motional states. The authors still do not state and justify which likelihood was used.

We thank the reviewer for the comment. To determine the number (N) of motional states, we used Akaike information criterion (AIC) (Eq. 11), in which the first term of Eq. 11 is the log-likelihood. Because the model with the minimum AIC value is selected as the preferred model, the calculation of AIC values requires a maximum likelihood estimation (MLE) of the parameters in each N -state model. We describe our use of the AIC and MLE to estimate the motional states on page 23-24, lines 511-540 in the revised manuscript.

389-393. Confusing language. On line 389 "n" is the cluster size. In line 392 the authors appear to say $\sqrt{n} C_n \sigma$ is the size.

We have corrected our description of $\sqrt{n} C_n \sigma$ to “the number of n -mer clusters” on page 24 (lines 547-550) in the revised manuscript.

295: what is the "value" of the peak?

We replaced the ambiguous word “value” with “brightness corresponding to the peaks”

295: do they mean the brightness histogram of the original image in the ROI or of the binarized image? I'm guessing they mean the former.

As the reviewer comments, the brightness histogram was obtained from the original image. We make this point clear on page 16 (lines 359-360) and Supplementary Fig. 11.

299: highest "value" of what? Distribution of the "values" of what? I could not work this out. I have no idea what the Gaussian function has been fitted to.

We agree with the reviewer about that the ambiguity of our expressions. We changed “highest value” and “distribution of the values” to “highest E (PFS₀)” and “sharpness distribution”, respectively. The Gaussian function was fitted to the sharpness distribution. Please see page 16-17 (lines 364-372) in the revised manuscript.

318: "The difference between..." This sentence is confusing. The difference between which quantity per pixel predicted by the network is compared to the correct value for that quantity per pixel?

To avoid confusion, we changed the term "error" to "average residual square (ARS)". Assuming the researcher's answer and the network output are defined as $d_{i,j}$ and $y_{i,j}$, respectively, for (i,j) pixel ($0 \leq i, j < 512$), then the error is,

$$ARS = \sqrt{\frac{1}{512^2} \sum_{i=0}^{511} \sum_{j=0}^{511} (y_{i,j} - d_{i,j})^2}. \quad (2)$$

We added this definition of ARS with the equation and explained the numerical meaning in "Optimization of the number of layers" of the Methods section in the revised manuscript on page 18. Reviewer #2, comment #1 makes a similar concern. Please see our response to that comment as well.

404-405: The authors should state in the text that they do a least squares fit as explained in their response.

Examples of confusing language (not exhaustive):

We thank the reviewer for the comment. We state the use of a least squares fit in "Cluster size estimation" of the Methods section in the revised manuscript (page 24-25). The fitting method was changed to MLE in the revised manuscript by performing the inverse variance weighted least squares. This method contained more information about the data distribution than LSE, thus providing a more precise fit.

90-93: Gives the impression that a problem with conventional systems is that they successfully acquire clear images. If I correctly understand their intended meaning, then a less confusing possibility is "The method successfully acquires clear single-molecule images for all observation fields, avoiding a substantial problem in conventional focus-keeping systems that comes from a difference in the refractive index between water and cells."

We thank the reviewer for the suggestion. We have changed the text accordingly on page 5, lines 93-95.

Minor language points I noticed (also not exhaustive):

303: "image in cells" - "images of cells"

307: "designed" - wrong word? Perhaps "chosen"?

308: "with applying Adam" - "using Adam"

311-312: "center of gravity" - "centroid"

We thank the reviewer for the comment. We have made changes in the revised manuscript as follows.

1. We changed “image in cells” to “images of cells” in line 376.
2. We removed the sentences using the word “designed”.
3. We replaced “with applying Adam” with “using Adam” in line 381.
4. We replaced “center of gravity” with “centroid” throughout.

Our response to Reviewer #2:

We are grateful to reviewer #2 for the critical and informative comments that have helped improve our paper. We write the comments made by the reviewer in ***bold italics*** with our responses below. Our changes in the manuscript are marked with highlighted **Green-colored font**.

Remaining major issues in the revised manuscript:

1) It is not clear how the error values in the machine learning procedures were defined. Are these incorrectly assigned pixels? Or totally missed targets?

This question could be addressed by a providing a training image data set (microscope image plus researcher generated/painted answer) together with deep learning generated answer images.

Let's say the cell in an image covers only 20 percent of all pixels, then how much of the cell area is incorrectly recognized given an error or 0.1? Is that 10 percent of all pixels in the image, or about half of the cell?

The revised manuscript falls short of clarifying the deep learning algorithm as requested by all three referees. The figures sketching the dimensionality reduction are insufficient.

We thank the reviewer for the comment. The term “error” in the previous manuscript was changed to “average residual square (ARS)” for clarity. We defined ARS as follows,

$$ARS = \sqrt{\frac{1}{512^2} \sum_{i=0}^{511} \sum_{j=0}^{511} (y_{i,j} - d_{i,j})^2}. \quad (2)$$

Here, $d_{i,j}$ is the value at pixel (i, j) ($0 \leq i, j < 512$) in the researcher generated/painted answer; $d_{i,j}$ adopts 1 for the researcher’s selected pixels (e.g. cell region) and 0 for the other pixels (background). $y_{i,j}$ is the value at pixel (i, j) in the predicted results by the neural network; $y_{i,j}$ adopts real number values between 1 and 0. Thus, ARS is a root mean of the squared difference between the values allocated to each pixel in the raw images and the predicted images by the neural network. When the network has not learned, ARS takes a value > 0.5 . During the learning, ARS approaches 0, but also depends on the difficulty of the tasks. We define ARS in “Optimization of the number of layers” of the Methods section on page 18 and Eq. 2.

Following the reviewer's suggestions, we show the raw images acquired by the microscope, the training images prepared/painted by the researcher for machine learning, and the answer images predicted by the neural network in Supplementary Figures 12b, 12d, 15d, and 15e, with output images from the learned network. The predicted areas in the bottom images in Supplementary Figures 12b, 12d, 15d, and 15e are almost the same as those in the training data (middle images), reflected in the low ARS values (~ 0.1). We show the ARS values in Supplementary Figures 14 and 15c, which confirmed more than 90% of pixels were correctly predicted.

In the case supposed by the reviewer (20% of all pixels covered by a cell) and error of 0.1 (10% of all pixels are incorrect in the network output), 10-30% (=20-10, 20+10) of all pixels could be recognized as covered by cell. When the output value of each pixel from the neural network is assumed to take 0 or 1, half of the cell is incorrectly recognized.

In addition to sketching the dimensionality reduction, we show in the figures the whole procedure of the automated imaging including the deep learning algorithm as flowcharts in the revised manuscript. We added Supplementary Fig. 9 for the automated imaging analysis by AiSIS, Supplementary Fig. 10 for the coarse autofocusing ("first auto-focusing" in the previous manuscript), Supplementary Fig. 11 for the high precision autofocusing ("second auto-focusing" in the previous manuscript), and Supplementary Fig. 12 for the cell searching/cell region detection in the revised manuscript. We also detail the deep learning algorithm of the Methods section (pages 17-18, lines 374-399). The figures sketching the dimensionality reduction are provided in new Supplementary Figs. 12 and 13 and original Supplementary Fig 2.

2) The description of the symbols used in Equation 2, the two-dimensional gaussian distribution, makes not much sense. You say that i and j are each between $-ROI$ and $+ROI$. Why do you subtract ROI from i and j in the gaussian? Why do you call $I_{\{i,j\}}$ the cross-correlation value at (i,j) ? Isn't it just the intensity of the gaussian? You said that you calculated the cross-correlation between the image and the two-dimensional gaussian. That's not what the equation shows. Very confusing!

We agree with the reviewer about the ambiguity of these expressions. Following the reviewer's comments, we changed Eq. 2 in the previous manuscript to Eq. 3 in the revised manuscript as follows,

$$I_{i,j} = \frac{1}{2\pi\sigma^2} \exp\left(-\frac{i^2 + j^2}{2\sigma^2}\right) \quad (3).$$

As the reviewer comments, $I_{i,j}$ is the intensity of Gaussian distribution. Here, (i, j) indicates the X-Y position in a ROI (Supplementary Fig 16a), and σ means standard deviation. i and j took values from -5 to 5 (pixels), and σ was set to be 2 pixels, which covers the whole single molecule spot.

We calculated the cross-correlation between the obtained raw image and the two-dimensional Gaussian (Eq. 3). The cross correlation is described as follows,

$$y_{i,j} = \frac{\sum_{I=-R}^R \sum_{J=-R}^R I_{I,J} x_{i+I, j+J}}{\sqrt{\sum_{I=-R}^R \sum_{J=-R}^R I_{I,J}^2} \sqrt{\sum_{I=-R}^R \sum_{J=-R}^R x_{i+I, j+J}^2}}, \quad (4)$$

where $x_{i,j}$ are pixel intensities at (i, j) in the obtained raw image, and $y_{i,j}$ are pixel intensities at (i, j) in the cross-correlated image. By binarization of $y_{i,j}$ with an appropriate threshold, the fluorescent spots were detected and labeled on the raw images. We carefully set this threshold as shown in new Supplementary Fig. 16. When the fluorescent spot was detected in the ROI, the centroid and intensity were determined by fitting the raw image with the following equation,

$$I(x, y; I_0, x_g, y_g, \sigma_A, a, b, I_{back}) = I_0 \exp\left(-\frac{(x-x_g)^2 + (y-y_g)^2}{2\sigma_A^2}\right) + a(x-x_g) + b(y-y_g) + I_{back}. \quad (5).$$

Here, I is the fluorescence intensity at the pixel position (x, y) . The intensity distribution is expressed as a Gaussian function with peak intensity of I_0 at the centroid, (x_g, y_g) and variance, σ_A^2 , on the background, which has inclinations of a and b for x and y directions, respectively, above the offset intensity, I_{back} . The fitted parameters are I_0 , x_g , y_g , σ_A^2 , a , b , and I_{back} . A single-molecule trajectory was obtained by connecting the centroids of the spots.

We added these descriptions in ‘‘Single-molecule tracking algorithm’’ of the Methods section (pages 21-23, lines 483-509) and added Supplementary Fig. 16 in the revised manuscript. We hope these additional descriptions are sufficient to resolve the concerns raised by the reviewer.

Our response to Reviewer #3:

We are grateful to reviewer #3 for the comments that have helped improve our manuscript. As indicated in the responses that follow, we have addressed the comments in the revised version of our paper. We put the comments made by the reviewer in ***bold italics*** with our responses below. Our changes in the manuscript are marked with highlighted **Green-colored font**.

I thank the authors for their detailed letter. My feeling is that the paper still falls short of a comprehensive explanation of the neural network approach and demonstration of its benefits. Notably, the description of the amount of data needed is very vague in the main text (e.g. the number of training data is mentioned as 1000 in the response letter but not mentioned in the text). I am still unconvinced that a manual annotation of 1000 images is a price worth paying for a method that will need to be redesigned for every new cell type. Could we quantify how much time/ effort is saved overall if we take into account preparation of the training set, selection of the model, etc, compared to a standard study? Additionally, I feel the description of some of the choices is still lacking, e.g. how to choose the number of layers is very poorly described, and the choice of activation function/ filters is not justified at all.

We thank the reviewer for the helpful comments to improve our manuscript. Based on the comments, we have made several amendments to the original manuscript with new Supplementary Figures.

To explain the neural network approach more clearly, we detail the deep-learning algorithm for coarse autofocusing and auto cell searching and the method for the auto cell region detection on pages 15-18, line 319-399, and we added several new figures that show flowcharts and schematics of the neural network algorithm (Supplementary Fig. 9 for automated imaging, Supplementary Fig. 10 for coarse autofocusing (“first auto-focusing” in the previous manuscript), and Supplementary Fig. 12 for cell searching/cell region detection) to help readers understand the neural network algorithm.

With regards to the number of training data required for the learning, we indicated the numbers in the main text as 400, 40, and 200 images for coarse autofocusing, cell searching, and cell region detection, respectively (lines 432-433) in the revised manuscript. We provide data that show the relationship between the number of training data and overlearning (error) in Supplementary Fig. 14, from which we

determined the appropriate number for the training. With these changes, we believe that the explanation of the neural network approach have become more convincing.

We agree with the reviewer about concerns over re-learning for every new cell type, because the SRIC images of cells may change between cell types. To see whether cell type-dependent learning is required for cell searching/cell region detection, we used CHO cells and HeLa cells, because these cell types exhibit distinct appearances from each other. As shown in new Supplementary Fig. 15, additional learning was not needed for cell searching/cell region detection unless the feature of the cell was visually obvious different. Neural networks that pre-learned with CHO cells were able to work well for HeLa cells and vice versa, although the accuracy was slightly lower. To show this point clearly in the revised manuscript, we added “Cell type-dependent learning” to the Methods section on page 20-21 and new Supplementary Fig. 15.

We report the time required for the manual preparation of the training images and the learning by the neural network in “Amount of training data” on pages 19-20, estimating it to be 160 and 646 minutes for automatic cell searching and cell region detection, respectively. In fact, we could establish a new setting for the automated functions through the time from the acquisition of the cell images to completion of the teaching process within a day. Therefore, we believe our deep learning approach will be preferred for several biological experiments. For example, image processing filters were automatically generated based on the easily prepared training data, whereas in conventional image processing, density processing, edge extraction, binarization, and other processes are done by trial and error and then combined seamlessly, making it difficult to predict the time and effort needed. We have included these points in the main text on lines 114-118 and in “Optimization of the number of layers” of the Methods on pages 18-19.

We added details on how to determine the number of layers and the corresponding equation for the error used in “Optimization of the number of layers” on page 18-19 in the revised manuscript. Reviewer #2, comment #1 makes a similar comment. Please see our response to that comment.

Concerning the choice of activation function/filters, we have modified “Optimization of the number of layers” on page 18-19 in the revised manuscript. In brief, we used the average residual square (ARS) to evaluate the neural network as described in our reply to Reviewer#2, Comment#1. We used ARS to assess the activation (or transfer) function. As a result, only for the last output layer in coarse

auto-focusing was the sigmoid function ($f(x) = 1/(1 + e^{-x})$) found to be superior to the Rectified Linear Unit (ReLU) function ($f(x) = x (x > 0), 0 (x < 0)$), which was used in the other layers. We show these functions in Supplementary Figs. 2, 10, 12, and 13.

With these changes, we believe that the benefits of our machine learning-based approach to automated single-molecule imaging analysis are more convincing.

REVIEWERS' COMMENTS:

Reviewer #1 (Remarks to the Author):

The manuscript has been revised and many issues addressed. The authors may want to consider the following points which I believe would help the readers to follow and understand the work:

Remaining issues:

- the use of the word "prelearned" is confusing (p. 5, line 86) "the similarity to the prelearned in-focus images was evaluated by roughly adjusting the focus position." what they mean is that they classified (or evaluated) the images with the trained network. Similar p6 line 107.
- 337ff : "The parameters of the neural network, such as the number and type of layers, activation functions, the convolution/deconvolution weight and bias values of each layer, were optimized during the learning" this is confusing. The weights and bias-offsets were learned by optimising the NN (using "Adam"), number of layers and activation function are optimised as described in "Optimization of the number of layers" by comparing the residuals for different (explicit) choices
- 379 "To prepare the training data for machine learning, cell regions with or without suitable fluorescent spot densities were manually set to 1 or 0, respectively ". Apparently some region boundary selection or region painting with a brush was done: p20 -445 "Subsequently, the suitable regions for the single-molecule imaging and cell adhesion area in the obtained images were manually painted"
- 391 This is badly expressed, probably they sorted the results by area size.
- 420 it would be interesting to know which functions they have tried and in which order they did the selection: number of layers first, activation function second? the other way around? all combinations?
- 473ff there is a clear difference between in the results depending on the combination of learning source and the (test?) sample and the results are better if source and test sample are aligned. However the best in case "Sample CHO" is "CHO+Hela x1.5" Furthermore: the claim that cell types could be classified seems plausible but the evidence is a bit thin. The main claim that training results do not depend on the cell type is sufficiently supported.
- 497: here is a manual step, the selected threshold might work for similar experiments only and there may be the need to readjust it (see also Suppl Fig 16b)
- 509 This is a bit simple
- "Equations" (8), (9), (13), (16) and p25 line 551: An equation contains a "=" sign. They should put one in.
- I didn't follow "Cluster size estimation"

Regarding post analysis:

- I don't think they explain if and how they register their two-colour data.
- In their analysis of the motional states they conclude from the AIC that there are 3 states. It is not clear from Supp Fig 6 how much more likely 3 states are than 2 or 4. It would be helpful to include the relative likelihood of 2, 3 and 4 states from the AIC to justify their conclusion.
- They did not really address the comment I made last time about their single molecule tracking algorithm. They say that their single molecule tracking algorithm has several advantages over other tracking algorithms, although theirs is actually much simpler than state-of-the-art algorithms. They do not state what the advantages are apart from convenience of integration into their system.
- They give values and error bars for diffusion rates in several places - do they come from the SD distribution motional state analysis or the MSD analysis? This is not always clear in the discussion and they do not state in either case how they calculate error bars.
- My comment previously about justifying their noise statistics in their various fitting methods has still not been addressed properly. They explain what they have done much better now, but in all cases where they assume Gaussian statistics they do not justify why this is acceptable.

391-394. The introduction of $V_i(A, x, y)$ is very confusing. I think they are just saying that the image acquisition of fields was done in descending order of the of their areas. If so, I think they should say something like this and not introduce the V_i term.

450-452. I don't understand what the sentence here means.

Something is wrong with their equation 4 for the cross correlation. It is missing "y" terms and they do not define R.

533-536 The explanation for this would be clearer if they state the function for the likelihood first, which is only apparent when getting to the AIC which contains it. However I think their method here is sound.

523 Is σ_i the standard error in the mean of the square deviations or the standard deviation of the square deviations or something else? Is this appropriate? Is the assumption of independent noise in each t_i bin for the fit acceptable?

550-551 The text does not contain what they say in their response, and I still do not follow what they really mean. How does the expression they give (in their response) tell us the cluster size? What to they mean by cluster size?

Minor language issues:

Main text line 365. SI line 93."smoothened" used in a several places should be "smoothed"

507 "at the background" - I think they mean "plus a background" or "added to a background" or similar

703. Figure 1d, "evaluation value" is still a meaningless term. I believe from the vastly improved description that I now understand this, and it is the "sharpness". If so, why not use this informative term for the axis label and in the text?

Reviewer #2 (Remarks to the Author):

The revision nicely addressed the remaining open questions. The additional illustrations of the algorithms are well done.

Reviewer #3 (Remarks to the Author):

I thank the authors for addressing my concerns, the description of the method is much clearer and the advantages better quantified. I'm happy for this to go through now.

Response to the reviewers' comments:

Our response to Reviewer #1:

We greatly thank reviewer #1 for the valuable suggestions that helped improve our paper. As described in the following responses, we have taken all comments into account in the revised version. The reviewer's comments are provided in ***bold italics***, and our responses are provided below. Our changes in the revised manuscript are indicated by the "tracked changes" feature of Microsoft Word. The expression of 'lines XX-XX, page XX' in our response means location of the changes in the manuscript in which all changes have been accepted. We submitted both versions of manuscript with and without the record of tracked changes.

The manuscript has been revised and many issues addressed. The authors may want to consider the following points which I believe would help the readers to follow and understand the work:

Remaining issues:

- the use of the word 'prelearned' is confusing (p. 5, line 86) the similarity to the prelearned in-focus images was evaluated by roughly adjusting the focus position. 'what they mean is that they classified (or evaluated) the images with the trained network. Similar p6 line 107.

We thank the reviewer for the helpful comment. We removed the confusing expression "prelearned" and clarified the sentence: "During coarse shifting of the objective position, the trained neural network judged the iris images to determine whether the iris was in focus or not." (lines 88-90, page 6) and "Since out-of-focus images can also be trained,..." (lines 105-106, page 6).

-337ff : The parameters of the neural network, such as the number and type of layers, activation functions, the convolution/deconvolution weight and bias values of each layer, were optimized during the learning this is confusing. The weights and bias-offsets were learned by optimising the NN (using Adam), number of layers and activation function are optimised as described in Optimization of the number of layers by comparing the residuals for different (explicit) choices

We thank the reviewer for this accurate comment. As the reviewer wrote, the number of layers and the activation function were determined beforehand and used during learning to optimize the weight and bias of each layer. We corrected the sentence to: "Two neural network parameters, the convolution/deconvolution weight and the bias values of each layer, were optimized by the learning method and saved in a binary file with other parameters, such as the numbers/types of layers and activation functions." (lines 366-369, page 17)

-379 To prepare the training data for machine learning, cell regions with or without suitable fluorescent spot densities were manually set to 1 or 0, respectively ."Apparently some region boundary selection or region painting with a brush was done: p20 -445 Subsequently, the suitable regions for the single-molecule imaging and cell adhesion area in the obtained images were manually painted"

We thank the reviewer for pointing out this issue. In the training process, the manually painted areas and other areas were set to 1 and 0, respectively. We corrected the ambiguous expression as follows, “cell regions with suitable fluorescent spot densities were manually painted, the pixel intensity was set to 1, and the other region was set to 0.” (lines 408-410, page 19)

-391 This is badly expressed, probably they sorted the results by area size.

We thank the reviewer for the helpful comment. The suitable regions were sorted by their area size, and image acquisition was executed starting from the region with the largest area. Thus, we changed the sentence to: “After cell searching was completed, image acquisition was performed according to the descending order of the area sizes.” (lines 421-422, page 20)

-420 it would be interesting to know which functions they have tried and in which order they did the selection: number of layers first, activation function second? the other way around? all combinations?

We thank the reviewer for indicating that these points should be clear. We optimized the number of layers and the weights of the convolution/deconvolution layers and then tried to find the most effective activation function. We described the procedure as follows: “When designing the neural network, the procedure was as follows: first, we started from the neural network with a minimum number of layers and weight; second, we increased these values until the ARS was saturated, indicating that the numbers were suitable; and finally, we tried different activation functions of the final layer to design a neural network with the highest speed and the lowest ARS.” (lines 455-459, page 21)

-473ff there is a clear difference between in the results depending on the combination of learning source and the (test?) sample and the results are better if source and test sample are aligned. However the best in case Sample CHO's CHO+Hela x1.5'Furthermore: the claim that cell types could be classified seems plausible but the evidence is a bit thin. The main claim that training results do not depend on the cell type is sufficiently supported.

We thank the reviewer for the critical comment. The precision of the network output depends not only on the training data quality but also on the number of network parameters. The network of

“CHO + Hela x1.5” includes 1.5 times the number of parameters than the other networks, thus providing higher precision. However, the focus here is not on the network structure but rather the effectiveness of training datasets for different types of cells. Thus, we removed the confusing data and the related description from Supplementary Figure 16 (Supplementary Figure 15 in the previous version) and the text. According to the obtained output, classification of the cell types seems possible but would depend on the differences in the cell-specific features. However, such a claim is difficult to support based on sufficient evidence, and we deleted the corresponding sentences.

-497: here is a manual step, the selected threshold might work for similar experiments only and there may be the need to readjust it (see also Suppl Fig 16b)

We thank the reviewer for the complementary comment. A typical situation in which the threshold has to be changed was described as follows: “This threshold should be adjusted depending on the signal-to-noise ratio of the single-molecule image (see also Supplementary Figure 17b).” (lines 523-525, page 24)

-509 This is a bit simple

We added a more detailed description of how to connect the fluorescence spots as follows, “To generate a single-molecule trajectory by connecting the spots, the following algorithm is used. All possible connections between two spots at times t and $t-1$ with a center-to-center distance below 6 pixels are listed. Then, the shortest connection between both spots is selected.” (lines 535-538, page 25)

- Equations (8), (9), (13), (16) and p25 line 551: An equation contains a '=' sign. They should put one in.

We thank the reviewer for pointing out this issue. We rewrote these equations with an “=” sign.

- I didn't follow Cluster size estimation"

An additional explanation about the EGFR cluster and its size estimation were provided in the revised manuscript. With these changes, we used the maximum likelihood estimation for cluster size estimation. (lines 129-130, page 7 and lines 584-598, page 28).

Regarding post analysis:

-I don't think they explain if and how they register their two-colour data.

We added explanations of the two-color data analysis at appropriate locations in the manuscript. (lines 216-217 and 219-221, page 11; lines 342-343, page 16; lines 580-582, page 28)

-In their analysis of the motional states they conclude from the AIC that there are 3 states. It is not clear from Supp Fig 6 how much more likely 3 states are than 2 or 4. It would be helpful to include the relative likelihood of 2, 3 and 4 states from the AIC to justify their conclusion.

We thank the reviewer for the helpful comment. We added a figure (Supplementary Figure 7b in the current manuscript) in which the sub-regions that include AIC values for states 2 to 4 in Supplementary Figure 7a are enlarged to show the state with the minimum value more clearly.

-They did not really address the comment I made last time about their single molecule tracking algorithm. They say that their single molecule tracking algorithm has several advantages over other tracking algorithms, although theirs is actually much simpler than state-of-the-art algorithms. They do not state what the advantages are apart from convenience of integration into their system.

We thank the reviewer for the comment. As the reviewer noted, our single-molecule tracking software was more convenient for incorporation into our automated system and could be modified according to the experimental purpose. However, we have not assessed any other advantages over another algorithm. Thus, we removed the confusing phrases.

-They give values and error bars for diffusion rates in several places - do they come from the SD distribution motional state analysis or the MSD analysis? This is not always clear in the discussion and they do not state in either case how they calculate error bars.

We thank the reviewer for the important comment. The diffusion coefficients and MSD were averaged for every cell, and the error bars indicate the SD (or SE in Figure 2a) calculated between cells. In addition, concerning the analysis of fluorescence intensity, the average and errors for fluorescence intensity were calculated over individual molecules. We added explanations in appropriate locations. (lines 543-545, page 25 and lines 772-773, page 38)

-My comment previously about justifying their noise statistics in their various fitting methods has still not been addressed properly. They explain what they have done much better now, but in all cases where they assume Gaussian statistics they do not justify why this is acceptable.

We thank the reviewer for this critical comment. In the previous manuscript, we assumed that the data obeyed a normal distribution for the cell-to-cell variation of the MSD (Supplementary Figure 6a, upper panel). However, careful examination of the cell-to-cell variation of the MSD revealed that a log-normal distribution was more likely (Supplementary Figure 6a, lower panel). The parameters calculated assuming either distribution showed a higher likelihood in the case using the log-normal distribution. According to the results, although the difference in the likelihood was slight and the parameters obtained using both methods were nearly consistent, we adopted a log-

normal distribution for parameter estimation (Supplementary Tables 3, 4, and 5).

391-394. *The introduction of $V_i(A, x, y)$ is very confusing. I think they are just saying that the image acquisition of fields was done in descending order of the of their areas. If so, I think they should say something like this and not introduce the V_i term.*

We thank the reviewer for this helpful comment. We deleted the confusing expression “ V_i ” and simplified the sentences as follows: “image acquisition was performed according to the descending order of the area sizes.” (lines 421-422, page 20)

450-452. *I don't understand what the sentence here means.*

Something is wrong with their equation 4 for the cross correlation. It is missing y 'terms and they do not define R .

We thank the reviewer for this appropriate comment. The sentence was improved as follows: “we completed preparation of the training data and learning procedure within one day”. (lines 484-485, page 22) In equation (4), we added terms based on the differences between the intensity of each pixel and the average. R was defined as the length of a side of the square ROI. (line 522, page 24)

533-536 *The explanation for this would be clearer if they state the function for the likelihood first, which is only apparent when getting to the AIC which contains it. However I think their method here is sound.*

We thank the reviewer for this supportive comment. We improved the explanation of AIC by using a general expression of the likelihood function. (lines 576-579, page 27)

523 *Is σ_i the standard error in the mean of the square deviations or the standard deviation of the square deviations or something else? Is this appropriate? Is the assumption of independent noise in each t_i bin for the fit acceptable?*

We thank the reviewer for this substantial comment. σ_i is a fitting parameter depending on an assumed noise distribution. For example, in the case of Gaussian distribution, σ_i means the standard deviation of the MSD values obtained by maximum likelihood method. Because a physical model is not available for the time dependency of σ_i , we fitted σ_i at each time point (Supplementary Table 3).

550-551 *The text does not contain what they say in their response, and I still do not follow what they really mean. How does the expression they give (in their response) tell us the cluster size? What to they mean by cluster size?*

We thank the reviewer for this comment. We used “clusters” to express the dimers, trimers, tetramers, and n -mers of EGFR, in which the monomers of EGFR interact with each other and form

a “cluster” of receptors. When EGFR-GFPs form clusters, the fluorescence intensity of the spot increases proportionally to the size of the cluster. Because the intensity histogram of the EGFR-GFP monomer shows a single Gaussian distribution, the histogram of the mixture of various size clusters consists of the sum of Gaussian distributions with means and SDs corresponding to the cluster sizes ($n\mu$ and $\sigma\sqrt{n}$ for n -mer). Equation (12) was changed from that in the previous manuscript, and C_n means the fraction of n -mer. We changed the text corresponding to the current equation. (lines 585-589, page 28)

Minor language issues:

Main text line 365. SI line 93. \$smoothed\$ used in a several places should be \$smoothed\$

We thank the reviewer for pointing out and correcting this issue (line 394, page 18 in the main text and the caption of Supplementary Figure 12).

507 \$t\$ the background\$ - I think they mean \$plus a background\$ or \$added to a background\$ or similar

As the reviewer suggested, we changed the expression to “plus a background”. (line 534, page 25)

703. Figure 1d, \$evaluation value\$ is still a meaningless term. I believe from the vastly improved description that I now understand this, and it is the \$sharpness\$. If so, why not use this informative term for the axis label and in the text?

We thank the reviewer for this comment. The term “Evaluation value for sharpness” was used to label the horizontal axis of Figure 1d.

Our response to Reviewer #2:

The revision nicely addressed the remaining open questions. The additional illustrations of the algorithms are well done.

We greatly thank reviewer #2 for this supportive comment.

Our response to Reviewer #3:

I thank the authors for addressing my concerns, the description of the method is much clearer and the advantages better quantified. I'm happy for this to go through now.

We greatly thank reviewer #3 for this reassuring comment.